# Analysis on the control of the black tiger shrimp in the America from the perspective of international cooperation

Yuntao Bai[1], Ruidi Hu[2], Lan Wang[3]*, Delong Li[4]

1 Business School, Shandong Management University, Jinan, China, 2 School of Public Administration and Policy, Shandong University of Finance and Economics, Jinan, China, 3 Center of Emergency Management, Chongqing Academy of Governance, Chongqing, China, 4 School of Business Administration, Inner Mongolia University of Finance and Economics, Hohhot, China

* wanglan-8722@hotmail.com

**Data Availability Statement:** All relevant data are within the paper and its Supporting information files.

**Funding:** This work is financially supported by Social Science Planning Foundation of Chongqing

## Abstract

The invasive black tiger shrimp has caused serious ecological problems in the America. However, since it can be directly eaten or made into feed, it may be beneficial to other countries. In order to ensure ecological security, it is necessary to control the invasion of the black tiger shrimp through international cooperation. Common control modes of the black tiger shrimp include the introducing natural enemy mode, making feed mode and the "bringing to the table" mode. In order to derive the applicable scope of various control modes of the black tiger shrimp and provide suggestions for the security and sustainability of the ecological supply chain of the America and cooperative country, this article constructs three differential game models and compares and analyzes the equilibrium results obtained by the models. Finally, the study shows that the higher the price of feed and the price of black tiger shrimp, the greater the degree of control of the black tiger shrimp. If the price of the black tiger shrimp and the reputation of the America for controlling the black tiger shrimp are lower, the America can gain more benefits under the feed production mode. Otherwise, the America prefers to sell the black tiger shrimp directly, thus directly "bringing to the table". Compared with the feed production or "bringing to the table" mode, cooperative country prefer to control the black tiger shrimp flooding through the natural enemy introduction mode.

## 1. Introduction

The black tiger shrimp, also known as the jumbo shrimp, tiger prawn, and black spotted prawn, is not a native American species, but a saltwater shrimp native to the Indo-Pacific region. These shrimp are large and usually dark gray to black in color with distinctive black stripes or spots on their bodies. They live primarily in shallow sandy or mud-bottomed environments in the ocean, especially in mangrove forests and estuarine areas. Black tiger shrimp may be found as an invasive species in the United States. In some cases, they may be found in waters along the southern coast of the United States, such as some waters in Louisiana and

in China (2021BS080); This work is financially supported by National Natural Science Foundation of China (72304157). The funders had no role in study design, data collection and analysis, decision to publish, or preparation of the manuscript.

**Competing interests:** The authors have declared that no competing interests exist.

Texas [1]. However, due to its strong reproductive capacity and adaptability, it has become one of the invasive species worldwide. The spread of black tiger shrimp is mainly caused by the following reasons. First, strong reproductive capacity. Black tiger shrimp has an extremely strong reproductive capacity, with a female shrimp able to lay hundreds to thousands of eggs annually. This high reproductive capacity makes its population grow rapidly and difficult to control. Second, strong adaptability. Black tiger shrimp has high adaptability to different water environments, and can survive and reproduce in various waters such as freshwater lakes, ponds, rivers, etc. They can tolerate a wide range of water quality conditions and temperature changes, so it is difficult to limit the expansion of the population through environmental factors [2]. Third, black tiger shrimp are omnivorous animals, which will feed on aquatic animals, plants, organic waste, etc., leading to competition and destruction of local species and ecosystems.

If an area adapts to the existence of invasive shrimp, it will spread rapidly and threaten the local ecological environment [3]. Similarly, the invasion of black tiger shrimp in the United States has a series of impacts on the ecological environment. First, the destruction of ecological balance. Black tiger shrimp is an alien species, which has no natural enemies and can reproduce rapidly. Its invasion will lead to the destruction of the local ecological balance. A large number of black tiger shrimp devour the eggs of benthic organisms and fish, resulting in the decrease of biodiversity in local waters. Second, the deterioration of water quality. The large number of black tiger shrimp will lead to the increase of organic matter and waste in water, resulting in the deterioration of water quality. They will disturb the bottom sediment, increase suspended substances, cause water turbidity, and affect the growth of aquatic plants. Third, competition and resource competition. Black tiger shrimp are highly competitive, and they compete for food, nests and habitats in water, resulting in increased competitive pressure on local species [4]. The invasion of black tiger shrimp will also lead to the exhaustion of local fishery resources, which has a negative impact on the livelihood and economy of local fishermen. Fourth, the ripple effect of ecological chain. The invasion of black tiger shrimp will have a ripple effect on the whole ecosystem. Local fish and other animals and plants that depend on the water habitat are threatened, which may trigger the collapse of the whole food chain and have a serious impact on the stability of the ecosystem. Fifth, damage to infrastructure. Due to the crawling ability of black tiger shrimp, they can penetrate into the crevices of water conservancy projects, ports and ships, causing damage to infrastructure. This not only poses a threat to the sustainable development of regional economy and infrastructure, but also increases the cost of maintenance and repair. In summary, the Local fish and other animals and plants that depend on the water habitat are threatened, which may trigger the collapse of the whole food chain and have a serious impact on the stability of the ecosystem. Fifth, damage to infrastructure. Due to the crawling ability of black tiger shrimp, they can penetrate into the crevices of water conservancy projects, ports and ships, causing damage to infrastructure. This not only poses a threat to the sustainable development of regional economy and infrastructure, but also increases the cost of maintenance and repair. In summary, the invasion of black tiger shrimp in the United States has caused serious negative impacts on the ecological environment. In order to protect the local ecosystem and promote sustainable development, it is necessary to take measures to control and manage the invasion of black tiger shrimp. This involves comprehensive management measures, including physical control, biological control, monitoring and appropriate regulatory measures.

International cooperation can be used to strengthen the control of the tiger shrimp invasion in the United States. Here are some possible modes. First, introduce natural predators. There are few natural predators of tiger shrimp in the United States, and they need to be introduced from cooperative countries. Before introducing a natural predator, sufficient research and

evaluation should be conducted to ensure that the species introduced will be effective against the tiger shrimp and will not cause new problems to the local ecosystem. The research should include the ecological characteristics, feeding habits, reproductive ability and potential impact on other species of the natural predator. Then, look for and select natural predators that are compatible with the tiger shrimp. These predators should be highly efficient predators and adapted to the local environmental conditions. Possible candidates include fish, crustaceans, insects or parasites. Natural predators can reproduce themselves, which can double the effect of the control of the tiger shrimp. Second, make tiger shrimp into feed. Making tiger shrimp into feed is a way to use resources, reduce the tiger shrimp invasion and provide an alternative feed source for the aquaculture industry. However, market demand, processing technology and impact on the environment and ecosystem should be considered before implementation. Taking into account the above factors and working with relevant stakeholders, the use of black tiger shrimp feed can be effective in controlling the black tiger shrimp invasion. Third, put the black tiger shrimp on the table. Controlling the black tiger shrimp invasion by using the black tiger shrimp as food can promote its commercial value and sustainable use, while reducing its impact on the local water ecosystem. However, attention should be paid to compliance, food safety and sustainable use of resources in the implementation process to ensure the effectiveness and feasibility of the control measures.

With the deepening of research on black tiger shrimp, a large number of research results have emerged. Some scholars have analyzed the factors influencing the population of invasive shrimp. Muñoz et al. [5] analyzed the impact of bird migration routes on invasive shrimp. Schoolmann and Arndt [2] analyzed the dynamic characteristics of invasive shrimp populations. Vázquez et al. [6] analyzed the impact of temperature and salinity on the survival and development of invasive shrimp larvae. You et al. [7] used microsatellite and mitochondrial haplotype diversity to reveal the differentiation of black tiger shrimp populations. These studies involved animal activities, temperature, salinity and other external environments.

Some scholars have studied the impact of invasive shrimps on local ecosystems. For example, Candolin et al. [8] studied the ecological impact of environmental disturbances on invasive shrimps. Dodd et al. [9] predicted the impact of invasive "killer shrimps" on European freshwater ecosystems. Macneil et al. [10] analyzed the impact of invasive killer shrimps on large invertebrates. These studies involve the impact of invasive shrimps on ecosystems and other animals.

In terms of the management of invasive aquatic animals in the United States, the management of Asian carp has been studied more. For example, Kokotovich and Andow [11] suggested that the Asian carp invasion could be addressed through social consensus and new management methods. Battaglin et al. [12] studied how bioactive chemicals affect the distribution of Asian carp in rivers. Zielinski and Sorensen [13] studied the use of bubble curtain to change the transfer of Asian carp. These studies covered the main theories of the use of management, ecology, and physics to manage the spread of Asian carp.

There is little or no research on the management of black tiger shrimps. Some scholars only study how to manage invasive shrimps. For example, Fatemeh et al. [14] believed that invasive Japanese river shrimps could be used as a potential commercial fishing resource. However, black tiger shrimps differ from other types of invasive shrimps in ecological impact and economic impact. Black tiger shrimps usually have a negative impact on local water bodies and biodiversity, and are known for their encroachment, competition for resources and other behaviors. Other invasive shrimp species may also have similar impacts on local ecosystems, such as changing water quality, destroying vegetation and so on. Black tiger shrimps are considered as an economic resource in some regions, which can be used for food or aquaculture, but they may also have a negative impact on local fishery and agriculture. The economic

impact of other invasive shrimp species may be related to their role and quantity in local ecosystems. In summary, different invasive shrimp species have certain differences in species characteristics, places of origin, ecological impact and economic impact. Understanding these differences can help to develop corresponding management and control strategies to reduce the negative impact of invasive shrimps on local ecosystems and economies.

In order to make up for the deficiencies of the above research, this article proposed three modes of controlling the flooding of black tiger shrimp from the perspective of cooperation and compared them. The main innovation or contribution of this article is two.

Firstly, this article proposes three modes of controlling the invasion of black tiger shrimp, namely introducing natural enemy, making feed and "bringing to the table". There are few studies on the control of the invasion of black tiger shrimp in the United States, and they often analyze the effect of a single control mode. This article refers to the control modes of other invasive species, proposes three modes of controlling the invasion of black tiger shrimp, and compares and analyzes each mode, so as to provide a reference for relevant decision-makers to control the invasion of black tiger shrimp.

Secondly, this article analyzes from the perspective of cooperation. In the United States, black tiger shrimp may have a negative impact on the local ecological environment. However, the situation may be different in other countries. Black tiger shrimp is regarded as a delicious seafood that can be eaten in some Asian countries, especially China. They are widely cultivated and caught in these countries, contributing to the local economy and food supply. Therefore, it can be said that black tiger shrimp is a problem in the United States, but is regarded as a valuable resource in other countries. To manage the impact of black tiger shrimp, international cooperation and information sharing should be strengthened, scientific assessment should be carried out, and appropriate management measures should be taken to balance its economic and ecological impacts.

## 2. Methodology

Differential game theory is commonly used to analyze dynamic strategic decision-making problems, especially when multiple players make decisions in continuous time and their actions are interdependent. In analyzing the problem of black tiger shrimp (an invasive alien species) infestation in the United States, it can be modeled as a differential game in which different players (e.g., government, fishermen, environmental organizations, etc.) may have different objectives. The following is an overview of the analytical steps. First, define the players. Identify all relevant decision makers, which may include environmental authorities, fisheries, local communities and other environmental groups. Second, set constraints. Social, economic, and environmental factors may impose constraints on participants' decisions. This may include legal constraints, budgetary constraints, fishing season or area restrictions, etc. Third, consider control variables. Introduce control variables for each participant that affect the black tiger shrimp population. Control variables could be fishing effort, money or resources invested in control measures, etc. Fourth, create an objective function. Define an objective function for each participant. For example, an environmental organization may want to minimize the black tiger shrimp population to protect the native ecosystem, while the fishing industry may want to maximize its revenue, even if it means keeping the black tiger shrimp population high in order to catch them [15]. Fifth, construct a dynamic system model. Create a dynamic model representing the growth of the black tiger shrimp population, which can be a time-based differential equation that takes into account natural growth, fishing mortality, and other factors that affect population size. Sixth, solve for the Nash equilibrium. Use tools from differential game theory, such as the Hamilton-Jacobi-Bellman equation (HJB equation), to find the

optimal strategy for each participant, thus determining a Nash equilibrium, the point at which no participant has an incentive to unilaterally change his or her strategy. Seventh, perform sensitivity analysis. Analyze the sensitivity of the model to parameter changes to understand which factors are most critical to controlling the black tiger shrimp problem. Eighth, develop management strategies. Based on the results of the above analysis, recommend effective management strategies. These strategies should consider the goals of all participants and seek to strike a balance between ecological conservation and economic benefits [16].

## 2.1 Problem description, hypothesis, and variable definition

**2.1.1 Problem description.** Controlling the spread of black tiger shrimp in the America is of great significance to protect ecosystems, safeguard economic interests, prevent environmental disasters and protect human health. It is a key measure to promote sustainable development and build a healthy society. This article selects the America and its cooperative country as game players. The America is game player 1 and the cooperative country is game player 2. In the process of cooperation between the United States and cooperating countries in the management of black tiger shrimp, policies can be developed, quarantine measures can be implemented, and public education can be promoted. Decisions made in one country have an impact on decisions made in another. For this reason, game theory dealing with conflict and cooperation can be an effective solution to this problem.

The task of controlling the American black tiger shrimp is long-term, uninterrupted and continuous. This is mainly due to the following reasons. First, strong reproductive ability. The black tiger shrimp has a strong reproductive ability, a pair of black tiger shrimp can produce a large number of eggs, and can quickly adapt and reproduce under various environmental conditions. This means that even after a control of the black tiger shrimp, if there is no continuous monitoring and control, the population is easy to re-spread, causing new problems. Second, difficult to completely eliminate. The black tiger shrimp is difficult to completely eliminate, because they have multiple survival stages in water bodies, and can spread through artificial waterways and other ways. Even if success is achieved in a specific area, regular inspection and control measures are still needed to prevent the re-invasion. Third, prolonged life cycle. The black tiger shrimp has a relatively long life cycle, which can last for several years. This means that control measures need to be sustained and across multiple life cycles to capture and control the black tiger shrimp at different stages. Fourth, long-term and uninterrupted prevention measures. The black tiger shrimp can be re-introduced through transnational trade, introduction of aquatic species, artificial release, etc. [2]. Therefore, it is necessary to continuously strengthen quarantine and control at borders and entry ports to prevent the introduction of new populations of the black tiger shrimp into the country. Therefore, in order to effectively manage the black tiger shrimp problem in the United States, it is necessary to continuously invest resources and efforts, take long-term, uninterrupted, continuous management measures and strengthen prevention work, so as to maintain the stability of the ecosystem and protect the local economic and social interests. In order to solve this long-term, uninterrupted, continuous problem, this article uses differential game, a time-continuous game.

Specifically, the United States and its cooperative country have three main approaches to the black tiger shrimp problem.

(1) Introducing natural enemy. In response to the tiger shrimp invasion in the United States, some natural enemy have been introduced to control the population of tiger shrimp. Here are some natural enemy that may be introduced. First, microctonus spp. It is a parasitic wasp that feeds on tiger shrimp. They lay eggs in the shrimp, and the host adults hatch and parasitize the shrimp, limiting the ability of the shrimp to reproduce. Second, cichlasoma

urophthalmus. It is a predatory fish that has been introduced to the waters to prey on tiger shrimp. They have a predatory pressure on both larvae and adults of tiger shrimp, effectively controlling the population. Third, procambarus clarkii. It is a reptile that loves to eat tiger shrimp. They mainly prey on tiger shrimp in wetlands and water bodies, effectively controlling the population. However, this approach is itself controversial for several reasons. First, unpredictable ecological consequences. The introduction of a new species as a control measure may have unintended ecological effects. The new species may turn out to be new invasive species and cause more damage to the native ecosystem. Second, food web disturbance. Introduced natural enemies may prey on species other than the target species and may even disrupt the balance of the original food chain. Third, adaptability problems. Introduced species may not be able to adapt to the new environment, or may adapt so well that their populations get out of control. For the control of black tiger shrimp, in addition to considering the introduction of natural enemies, other safer methods such as physical harvesting, habitat management, chemical control, public education and exotic species reporting system should be considered in a comprehensive manner. The introduction of natural enemy is a complex control strategy that requires scientific evaluation and management. Before the introduction of any species, sufficient research and experiments must be conducted to ensure that the introduction of the species will not pose a greater threat to the local ecosystem and can effectively control the population of tiger shrimp.

(2) Feeding mode. The conversion of American black tiger shrimp into feed is a possible strategy to control the tiger shrimp invasion. This approach can be applied in fisheries and aquaculture to utilize the black tiger shrimp resources and reduce its impact on native species and ecosystems. However, this strategy also needs to be evaluated and managed comprehensively to ensure its feasibility and effectiveness and avoid introducing other potential problems. For example, in the process of addressing the Asian carp invasion, Bowzer et al. [17] proposed the conversion of Asian carp into feed to address the ecological problems. The following are some factors to be considered. First, safety. Ensure that the use of black tiger shrimp as feed does not have a negative impact on the target species or human health. A safety assessment must be conducted, including checking for the existence of possible toxicity or pathogenicity, and ensuring that the black tiger shrimp is properly handled and treated. Second, sustainability. The sustainable supply of black tiger shrimp as feed should be ensured. This means that the renewability of the black tiger shrimp resources, the feasibility of aquaculture and capture, should be evaluated to avoid overfishing or resource depletion. Third, ecological impact. Evaluate the potential impact of black tiger shrimp as feed on local ecosystems. Although this method can control the population of black tiger shrimp, the introduction of more species may bring other changes to the local ecosystem. Therefore, a comprehensive analysis must be carried out to ensure that the control measures do not have a negative impact on the sustainability and stability of the overall ecological balance. Below are some considerations on the ecological feasibility and long-term sustainability of the feeding mode. First, the ecological feasibility is analyzed. Firstly, the proliferation of black tiger prawns may disrupt the local ecological balance, and harvesting them and converting them into feed may help to minimize their negative impact on the environment. Second, by harvesting and utilizing black tiger prawns as forage, their populations can be effectively controlled, reducing pressure from competing native species for food and habitat. Long-term sustainability is then analyzed. First, continued demand ensures the sustainability of the commercial fishery and the manufacturing industry can generate profits from the sale of black tiger shrimp feed. Second, effective management systems must be established and maintained to ensure that the black tiger prawn fishery is conducted in a sustainable manner without causing new ecological problems. Thirdly, the process of producing black tiger shrimp as feed should be as efficient as possible to minimize energy

consumption and waste generation and to improve the environmental friendliness of the production. In summary, the production of American black tiger shrimp as feed is a potential control strategy, but it needs to be comprehensively evaluated and managed to ensure its safety, sustainability and ecological feasibility. This requires close cooperation and scientific guidance between relevant departments, scientific research institutions and stakeholders.

(3) "Bring to the table" mode. A potential approach to address the problem of black tiger shrimp invasion is to consume them. This has several advantages. First, sustainable use. By consuming black tiger shrimp, they can be used as a resource and reduce their impact on local ecosystems. This can transform black tiger shrimp from a problem species into a sustainable food for human consumption. Second, economic benefits. The consumption of black tiger shrimp can create economic value and promote the development of fishing and aquaculture. It provides a new source of income for fishermen and farmers, while stimulating the development of related industrial chains [14]. Third, enriching diet and culture. As a delicacy, black tiger shrimp can enrich people's dietary choices and bring new delicious experiences. At the same time, it can also become part of local food culture, adding local characteristics and attraction. However, this mode also needs to pay attention to the following issues. First, ecological impact. Although the consumption of black tiger shrimp is beneficial to control its population, it still needs to pay attention to its impact on local ecosystems. The residue treatment, food source, and aquaculture methods need to be properly managed to avoid environmental pollution and ecological damage. Second, reproduction control. In order to keep the population of black tiger shrimp under control, effective reproductive control measures are needed. Monitoring and managing the reproductive capacity of black tiger shrimp will ensure the reasonable control of the population. Third, food safety. It is very important to ensure the safety of eating black tiger shrimp. Food safety standards and regulations must be followed to ensure that the black tiger shrimp meets the requirements of physical examination, and appropriate cooking and processing methods are adopted. The ecological feasibility and long-term sustainability of this model are analyzed below. It is first analyzed in terms of ecological feasibility. First, encouraging the harvesting of black tiger prawns for food will help to control their overpopulation and reduce their pressure on the ecosystem. Secondly, by reducing the number of black tiger prawns, local marine biodiversity threatened by this invasive species can be protected. It is then analyzed in terms of long-term sustainability. Firstly, if black tiger prawns become popular as a delicacy, it could provide an ongoing economic incentive for harvesters and the restaurant industry to promote sustainable harvesting. Second, as awareness of sustainable food choices increases, black tiger prawns may become a food item that grows in demand. In summary, eating black tiger shrimp to solve the problem of their flooding is a potential approach to achieve sustainable use, economic development and enriching the food culture. However, ensuring their reasonable management, ecological friendliness and food safety is very important and requires the joint efforts of many parties, including governments, fisheries departments, farmers and consumers.

Among the three control modes of black tiger shrimp, if we adopt the strategy of introducing natural enemies of black tiger shrimp, then we can control the population of black tiger shrimp through natural predatory behavior, and there is no need for human supervision and management, which is more economical and environmentally friendly. However, the introduction of new species may bring other ecological risks, for example, the new species may become invasive in other ecosystems. At the same time, this method may disrupt local biodiversity and ecological balance. If the model of making black tiger prawns into feed is adopted, then not only can the population of black tiger prawns be reduced, but also the problem can be turned into a resource. Because black tiger prawns are rich in protein, they can provide a high quality protein source for livestock and poultry. However, manufacturing black tiger prawns

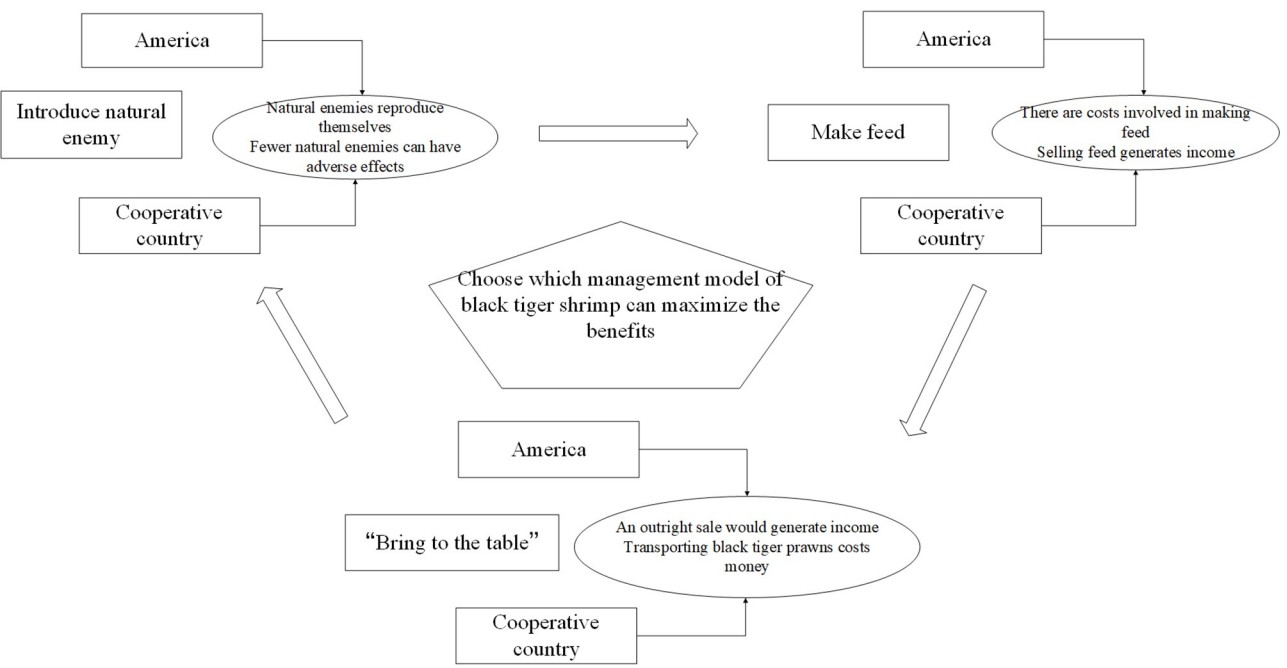

**Fig 1. Relationship between these three control modes of black tiger shrimp infestation.**

into feed requires the establishment of an effective harvesting, processing and marketing system, which can be costly. Also, black tiger prawns may pose a risk of disease if they contain microorganisms or viruses that are harmful to livestock growth. If the model of direct consumption of black tiger prawns is adopted, then market demand can be utilized for the purpose of shrimp flood control, opening up new markets and business opportunities. This is mainly because black tiger prawns have high nutritional and culinary value as seafood [18]. However, developing the demand for black tiger prawns, which may involve market research, promotion, and culinary development, requires a relatively large investment of resources and time. Over-exploitation may forget the original purpose of conservation and lead to overfishing of black tiger prawns instead. Overall, each approach has its own advantages and disadvantages and needs to be chosen in the context of the specific environment and actual situation, as well as its combined environmental, economic and social impacts.

The relationship between the three control modes of black tiger shrimp infestation is shown in Fig 1.

**2.1.2 Hypothesis.** (1) The introduced natural enemy will not have a negative impact on the local ecology in the United States.

The natural enemy of the black tiger shrimp are mainly as follows: First, fish. Many fishes are capable of eating the black tiger shrimp, including carp, herring, trout, etc. Second, birds. Waterfowl such as herons, cormorants and other birds also eat the black tiger shrimp. Third, reptiles. Some reptiles such as water snakes and crocodiles may also eat the black tiger shrimp. Fourth, crustaceans. Some large crustaceans such as lobsters, large crabs, etc. are also natural enemy of the black tiger shrimp. Fifth, large aquatic insects. Some large aquatic insects such as water crustaceans have also been found to eat the black tiger shrimp. Introduction of natural enemy may pose a risk of damage to the local ecosystem, depending on the type of introduced natural enemy and environmental conditions. The purpose of introducing natural enemy is to

control the population of invasive species (such as the black tiger shrimp) to reduce its impact on the local ecosystem. However, introduced natural enemy may have negative effects on local species, such as predation of native rare species, ecological balance damage, etc. Black tiger shrimp have a variety of natural enemies and whether or not they harm the ecosystem depends on the characteristics of the ecosystem itself. If appropriate natural enemies are introduced based on the local ecosystem in the United States, then there will be no impact on the local ecosystem [19]. When considering the introduction of natural enemy, a full scientific assessment and management plan must be carried out. This includes assessing the risks and benefits of introducing natural enemy, whether they will adapt to the local environment, whether they will exceed the control range of the target species, etc. At the same time, strict monitoring and control measures should be taken to ensure that the introduction of natural enemy is limited and avoids further damage to the local ecosystem. Therefore, before introducing natural enemy, careful weighing of the pros and cons and sufficient scientific evaluation should be carried out to ensure that the potential threat to the local ecosystem is minimized.

(2) The decline of a species can have an impact on the ecology of the America partner country.

The decrease or increase of a species may have an impact on the local ecosystem [20]. Each species plays a specific role and function in the ecosystem, which is called niche. Species interact with each other and depend on each other to maintain ecological balance. When a species decreases or disappears, it may have a knock-on effect on the ecosystem. First, the destruction of the food chain. The decrease of a species may lead to a break in the food chain. It may be the food source of other species, and if it decreases, it will cause a food shortage in the next level of species. Second, the breakdown of ecological balance. Species interact with each other, such as predation, competition, symbiosis, etc. When a species decreases, these interactions may be broken, leading to the imbalance of ecological balance. Third, the loss of ecological functions. Each species participates in ecological processes in different ways, such as pollination, seed dispersal, food decomposition, etc. The decrease of a species may lead to the loss or impact of these ecological functions. Fourth, the decrease of species diversity. The decrease of a species means the decrease of species diversity. Species diversity plays an important role in the stability and resilience of ecosystems, so the disappearance of a species may reduce the ability of an ecosystem to resist external disturbances. Therefore, maintaining species diversity and balance in an ecosystem is crucial to ensure its health and stability. The decline of a species may disrupt this balance and have a negative impact on the local ecosystem.

(3) Feed logistics costs are ignored.

If the cost of feed logistics is ignored, it will have a certain impact on agricultural production and supply chain management. The cost of feed logistics refers to the cost of transporting feed from the production site to the farm or farm, including transportation costs, storage costs, insurance costs, etc. For agricultural production and supply chain management, it is important to reasonably consider and manage the cost of feed logistics, which can help agricultural producers and supply chain managers make decisions on economic and environmental sustainability. The cost of feed transportation is composed of many factors and components, so it can be regarded as a complex problem. It is composed of the cost of transportation distance, transportation mode, transportation capacity, loading efficiency, fuel and energy, etc. Therefore, the cost of feed transportation needs to consider the above factors comprehensively, and evaluate and manage according to the specific situation. Of course, if transportation costs are ignored in the process of managing the black tiger prawn infestation problem, the following implications may arise. First, resources are distributed unequally. If the management strategy does not take into account the transportation costs required to actually respond to black tiger shrimp in each region, it may result in some regions receiving insufficient resources to

cover the costs of transporting and treating the invasive species. As a result, control in these areas may be less effective than in others. Second, it is not cost-effective. Treatment strategies usually require a large investment, and if transportation costs are not taken into account when developing the strategy, this may result in a lower overall input-output ratio and a less cost-effective decision. Third, strategy implementation becomes more difficult. Transportation costs can affect the feasibility and efficiency of management measures, and if they are not properly accounted for in budgets or strategies, they may encounter insufficient funds to implement effective control measures. This article mainly studies the ecological problem of how to control the flood of black tiger shrimp, rather than the logistics problem. For convenience, this article assumes that the cost of feed logistics is ignored.

**2.1.3 Variable definition.** There are a variety of ecological, economic, and socio-behavioral factors that need to be considered in the management of black tiger shrimp (invasive species) through modeling. The following are some of the key factors that should be considered when selecting variables for modeling. First, ecological factors, which primarily include the reproductive rate of black tiger shrimp. These parameters are critical for understanding population dynamics. Second, economics factors. For example, the economic value of black tiger shrimp to local fisheries and the cost of harvesting; the input costs of managing and controlling black tiger shrimp infestations; and the potential impacts of black tiger shrimp infestation on other economic activities. Third, socio-behavioral factors. These include, among others, local community attitudes: perceptions of black tiger shrimp and the reputation gained for control measures.

When constructing the differential game model in this article, many parameters and variables are designed. These parameters and variables are defined as shown in Table 1.

The process of combating invasive species, such as black tiger shrimp, in international cooperation is not directly quantifiable in terms of "reputation", as reputation is a relatively subjective and complex concept that is often closely related to a country's demonstrated behavior, commitment and international image over time. Indirectly, however, the reputation of a country or multinational cooperation can be assessed and perceived in the following ways. First, the degree of participation. The level of active engagement demonstrated by participating countries in international cooperation, including financial contributions, technical support, human investment, etc., can reflect the sense of responsibility and leadership of these countries [21]. Second, effectiveness of results. The effectiveness of governance measures, such as reducing the black tiger shrimp population and restoring ecological balance, is an important aspect of evaluating the reputation of participating countries. These effects can usually be demonstrated through scientific studies and monitoring data. Third, international commendations and awards. Recognition and awards through international organizations such as the United Nations can also reflect the achievements and positive impacts of a country or multinational cooperation in the governance process. Fourth, media coverage and public perception. How international and national media report on the process and results of cooperation, and the public perception shaped by these reports, are also important ways of assessing reputation [22]. Fifth, environmental and sustainable development indicators. Various international environmental and sustainable development rankings or indicators can provide a reference to a country's environmental governance and international cooperation achievements. It is worth noting that, as each country's situation is different, the reputation achieved will manifest itself in different areas and is difficult to measure clearly with quantitative data. Often, the reputation of a country or organization in the international arena is the result of accumulation over time and is not based solely on a single event or project. In practice, the assessment of reputation usually relies on multiple sources of information and observation over time.

**Table 1. The main definition of variables and parameters in this article.**

| variables and parameters | specific meaning |
|---|---|
| Y = {B,M,E} | three modes of controlling the invasion of black tiger shrimp (introduce natural enemy, make feed, "bring to the table") |
| independent variable | |
| $G_{Y1}(t)$ | the level of control of black tiger shrimp invasion in the United States under the control mode $Y$ |
| $G_{Y2}(t)$ | the level of control of black tiger shrimp invasion in the cooperative country under the control mode $Y$ |
| $x_{Y1}(t)$ | the United States gained a reputation for tackling the black tiger shrimp epidemic under the control mode $Y$ |
| $x_{Y2}(t)$ | the cooperative country gained a reputation for tackling the black tiger shrimp epidemic under the control mode $Y$ |
| parameter | |
| $\rho$ | the discount rate that occurs over time, $0 \leq \rho \leq 1$ |
| $\delta$ | decay of reputation, $\delta > 0$ |
| $b_1, b_2$ | revenue per unit of black tiger shrimp managed by the United States or the cooperative country, $b_1, b_2 > 0$ |
| $c_1, c_2$ | cost of a unit of black tiger shrimp managed by the United States or a cooperative country, $c_1, c_2 > 0$ |
| $l$ | the positive impact of reputation per unit quantity, $l > 0$ |
| $c_B$ | negative impact on the ecology of the cooperative country reduced per unit number of natural enemy, $c_B > 0$ |
| $h_B$ | the reproductive rate of the natural enemy, $h_B > 0$ |
| $f_1, f_2$ | reputation for the quantity of black tiger shrimp controlled by the United States or the cooperative country, $f_1, f_2 > 0$ |
| $f_B$ | negative effect of reduced number of natural enemy per unit on the reputation of the cooperating country, $f_B > 0$ |
| $c_{M1}, c_{M2}$ | unit cost of feed production in the United States or the cooperative country, $c_{M1}, c_{M2} > 0$ |
| $q_{M1}, q_{M2}$ | selling price of feed in the United States or in the cooperative country, $q_{M1}, q_{M2} > 0$ |
| $c_T$ | unit transportation cost of black tiger shrimp, $c_T > 0$ |
| $f_E$ | satisfaction of eating a unit quantity of black tiger shrimp in the cooperative country, $f_E > 0$ |
| $q_{E1}, q_{E2}$ | the price of black tiger shrimp sold directly in the United States or in the cooperative country, $q_{E1}, q_{E2} > 0$ |
| function | |
| $J_{Y1}(t)$ | the social welfare function of the United States under the control mode $Y$ |
| $J_{Y2}(t)$ | the social welfare function of the cooperative country under the control mode $Y$ |
| $V_{Y1}(t)$ | the social benefits of the United States under the control mode $Y$ |
| $V_{Y2}(t)$ | the social benefits of the cooperative country under the control mode $Y$ |

## 2.2 Differential game of three control modes

Game theory is a mathematical theory that studies the behavior of interactions between decision makers (participants) with conflicting and cooperative motives. When dealing with invasive species, such as black tiger shrimp, multiple interested parties or countries may adopt different management strategies because they have different goals, resources, and attitudes. The use of game theoretic models to analyze and predict the behavior of these participants in the management of black tiger shrimp can provide multiple benefits. First, strategy design. Game theory can help determine the best strategy for each participant, i.e., how to cooperate with other countries to control the black tiger shrimp population in the ecosystem while protecting their own interests. Second, consideration of externalities. In such a model, participants

consider the impact of their behavior on other participants (positive or negative externalities). For example, a country may invest more in controlling black tiger shrimp, but this may also result in the shrimp escaping from that country into the waters of other countries. Third, identify barriers to cooperation. Game theory can reveal natural barriers that lead to ineffective governance or failed cooperation, such as inconsistency across policies or unequal distribution of costs and benefits. Fourth, incentive formulation. Designing incentives with the help of game theoretic models, such as international aid, subsidies, or fines, promotes cooperation among countries to jointly implement effective management strategies and mitigate ecological and economic impacts [15]. Fifth, analyzing dynamic interactions. Game theory allows analyzing the decision-making process of multiple participants over a time series to understand and predict the long-term dynamics of policy change resource management outcomes. Sixth, circumventing confrontation. Coordinated action is often required in resource management, and game theory helps to identify ways to avoid confrontation (e.g., the Prisoner's Dilemma) and promote mutually beneficial cooperation. Thus, the use of game theoretic models can help policy makers better understand the stakes and potential cooperation/confrontation scenarios of the various participants, and thus direct more effective international or regional cooperative management strategies to reduce the potential negative ecological and economic impacts of invasive species like the black tiger shrimp.

Differential games are a branch of game theory that deals with dynamic strategy decision problems in which the control variables of the participants change continuously over time and these changes affect the state of the system. When dealing with the management and governance of invasive species such as black tiger shrimp, choosing differential games has the following advantages. First, dynamic process modeling. Differential games can naturally describe and deal with time-dependent dynamic systems. The ecological process of species invasion is continuous and time-varying, and differential games can be used to model the dynamic process of shrimp population change over time and participants' adjustment of strategies. Second, optimal control. Through the differential game, participants can learn how to adjust resource allocation and develop preventive and control measures under changing circumstances (e.g., increase or decrease in black tiger shrimp populations), with the aim of achieving long-term management goals. Third, future impacts are considered. Differential games enable participants to calculate the potential impacts of current decisions on future states through a forward-looking perspective, thereby optimizing long-term gains. Fourth, state-dependent strategies. In differential games, participants' decision strategies can depend on the current state of the system. This allows decisions to manage black tiger shrimp to be flexibly adapted to real-time or near real-time ecosystem state changes [16]. Fifth, continuous time scenario analysis. Using differential games it is possible to simulate different management strategies of different participants in continuous time and explore various decision paths and their potential outcomes. Sixth, multi-party cooperation and disputes. Differential games help to model the collaboration and conflict between multiple interested parties on resource sharing issues, thereby promoting cooperative strategies for managing invasive species. Through differential game modeling, managers can more accurately predict and formulate strategies against black tiger shrimp invasions, making this approach particularly useful for complex decision-making problems in areas such as environmental management, resource use, and protection of ecological balance. Within the framework of game theory, differential games provide a powerful tool to help coordinate actions among participants with the ultimate goal of mitigating the possible negative impacts of the black tiger shrimp invasion. At present, the differential game it is mainly applied in the fields of advertising decision [23], logistics management [24], supply chain [25], etc.

If the US manages the tiger shrimp problem by introducing the natural enemy from the cooperative country, the US social welfare function consists of the ecological benefits of managing the tiger shrimp, the costs of managing the tiger shrimp, and the reputation gained from managing the tiger shrimp. The cooperative country's social welfare function consists of the ecological benefits of managing the tiger shrimp, the costs of managing the tiger shrimp, the reputation gained from managing the tiger shrimp, and the ecological impact of reducing the natural enemy on the cooperative country.

The social benefit function of the United States and its cooperative country are:

$$J_{B1} = \int_0^\infty \left[ b_1 G_{B1}(t) - \frac{c_1}{2} G_{B1}^2(t) + l x_{B1}(t) \right] e^{-\rho t} dt \tag{1}$$

$$J_{B2} = \int_0^\infty \left[ b_2 G_{B2}(t) + b_2 \ln(1+h_B) G_{B2}(t) - \frac{c_2 + c_B}{2} G_{B2}^2(t) + l x_{B2}(t) \right] e^{-\rho t} dt \tag{2}$$

In the above formula, $b_1 G_{B1}(t)$ represents the ecological benefit gained by the US in controlling the black tiger shrimp under the mode of introducing natural enemy. $\frac{c_1}{2} G_{B1}^2(t)$ represents the cost of controlling the black tiger shrimp in the US under the mode of introducing natural enemy. $l x_{B1}(t)$. represents the reputation gained by the US in controlling the black tiger shrimp under the mode of introducing natural enemy. $b_2 G_{B2}(t)$ represents the income gained by selling the black tiger shrimp's natural enemy under the mode of introducing natural enemy. $b_2 \ln(1+h_B) G_{B2}(t)$ represents the additional benefit brought by the breeding of the natural enemy to the ecology under the mode of introducing natural enemy. $\frac{c_B}{2} G_{B2}^2(t)$ represents the impact of the natural enemy' reduction on the ecology of the cooperative country under the mode of introducing natural enemy. $\frac{c_2}{2} G_{B2}^2(t)$ represents the cost of controlling the black tiger shrimp in the cooperative country under the mode of introducing natural enemy. $l x_{B2}(t)$ represents the reputation gained by the cooperative country in controlling the black tiger shrimp under the mode of introducing natural enemy.

The reputation gained by the government in the fight against the black tiger prawn infestation can be operationalized and measured in a number of ways. The success of the governance efforts is not only reflected in the ecological benefits, but also in the image and credibility that the government develops with the public. The following are some possible means of operationalization and measurement. First, public opinion survey. Public opinion survey to find out the public's perception and satisfaction with the government's handling of the black tiger prawn infestation problem. Second, media coverage. Analyze the coverage of related topics in news media, professional publications and social media to understand the image of the government in the media and the public. Third, ecological benefit assessment. Assessing ecosystem restoration and changes in black tiger shrimp populations, these scientific results can be used as an indicator of the effectiveness of the government's governance. Fourth, government performance assessment. Through in-depth government performance assessment reports, analyze the comparison between the actual effectiveness of governance and the original plan. Fifth, periodic reports. Publish periodic reports to demonstrate the progress and challenges of governance efforts and increase the public's and stakeholders' right to know. The ultimate goal of acquiring a reputation is to develop a good public image and gain public trust, which requires the government to demonstrate high performance not only in the planning and implementation of governance programs, but also in public relations and communication efforts. By considering the above operational and measurement tools together, the government is able to more accurately assess the reputation it has gained in its efforts to combat the black tiger prawn infestation. The change in the reputation of US and its cooperative country under the

mode of introducing natural enemy can be expressed as:

$$\dot{x}_{B1}(t) = f_1 G_{B1}(t) - \delta x_{B1}(t) \tag{3}$$

$$\dot{x}_{B2}(t) = (f_2 - f_B) G_{B2}(t) - \delta x_{B2}(t) \tag{4}$$

In the above formula, $f_1 G_{B1}(t)$ represents the increase in the reputation of the US for controlling the tiger shrimp invasion in the mode of introducing natural enemy. $\delta x_{B1}(t)$ represents the decline in the reputation of the US in the mode of introducing natural enemy. $f_2 G_{B2}(t)$ represents the increase in the reputation of the cooperative country for controlling the tiger shrimp invasion in the mode of introducing natural enemy. $f_B G_{B2}(t)$ represents the negative impact of the reduction of natural enemy on the reputation of the cooperative country in the mode of introducing natural enemy. $\delta x_{B2}(t)$ represents the decline in the reputation of the cooperative country in the mode of introducing natural enemy.

If all the tiger shrimp caught is used as feed, the social welfare function for the US and cooperative country in this mode are:

$$J_{M1} = \int_0^\infty \left[ b_1 G_{M1}(t) - \frac{(c_1 + c_{M1})}{2} G_{M1}^2(t) + q_{M1} G_{M1}(t) + l x_{M1}(t) \right] e^{-\rho t} dt \tag{5}$$

$$J_{M2} = \int_0^\infty \left[ q_{M2} G_{M2}(t) - \frac{c_{M2}}{2} G_{M2}^2(t) + l x_{M2}(t) \right] e^{-\rho t} dt \tag{6}$$

In the above formula, $b_1 G_{M1}(t)$ represents the ecological benefits of the US from controlling the black tiger shrimp invasion in the feed-making mode. $\frac{c_1}{2} G_{M1}^2(t)$ represents the cost of the US from controlling the black tiger shrimp invasion in the feed-making mode. $\frac{c_{M1}}{2} G_{M1}^2(t)$ represents the cost of the US from producing the feed in the feed-making mode. $q_{M1} G_{M1}(t)$ represents the income of the US from selling the feed in the feed-making mode. $l x_{M1}(t)$ represents the reputation of the US from controlling the black tiger shrimp in the feed-making mode. $q_{M2} G_{M2}(t)$ represents the income of the cooperative country from selling the feed. $\frac{c_{M2}}{2} G_{M2}^2(t)$ represents the cost of the cooperative country from producing the feed. $l x_{M2}(t)$ represents the reputation of the cooperative country from controlling the black tiger shrimp in the feed-making mode.

The change in the reputation of US and its cooperative country under the mode of making feed can be expressed as:

$$\dot{x}_{M1}(t) = f_1 G_{M1}(t) - \delta x_{M1}(t) \tag{7}$$

$$\dot{x}_{M2}(t) = f_2 G_{M2}(t) - \delta x_{M2}(t) \tag{8}$$

In the above formula, $f_1 G_{M1}(t)$ represents the increase in the reputation of the United States for controlling the black tiger shrimp invasion in the feed-making mode. $\delta x_{M1}(t)$ represents the decrease in the reputation of the United States in the feed-making mode. $f_2 G_{M2}(t)$ represents the increase in the reputation of the cooperative country for controlling the black tiger shrimp invasion in the feed-making model. $\delta x_{M2}(t)$ represents the decrease in the reputation of the cooperative country in the feed-making mode.

If the United States and its cooperative country choose to eat the black tiger shrimp to combat the invasion, the social welfare function for the US and cooperative country in this mode

are:

$$J_{E1} = \int_0^\infty \left[ b_1 G_{E1}(t) - \frac{c_1}{2} G_{E1}^2(t) + q_{E1} G_{E1}(t) + lx_{E1}(t) \right] e^{-\rho t} dt \tag{9}$$

$$J_{E2} = \int_0^\infty \left[ q_{E2} G_{E2}(t) - \frac{c_T}{2} G_{E2}^2(t) + lx_{E2}(t) \right] e^{-\rho t} dt \tag{10}$$

In the above formula, $b_1 G_{E1}(t)$ represents the ecological benefits of the US from controlling the black tiger shrimp flooding in the "bringing to table" mode. $\frac{c_1}{2} G_{E1}^2(t)$ represents the costs of the US from controlling the black tiger shrimp flooding in the "bringing to table" mode. $q_{E1} G_{E1}(t)$ represents the income of the US from directly selling the black tiger shrimp in the "bringing to table" mode. $lx_{E1}(t)$ represents the reputation of the US from controlling the black tiger shrimp in the "bringing to table" mode. $q_{E2} G_{E2}(t)$ represents the income of the cooperative country from directly selling the black tiger shrimp. $\frac{c_T}{2} G_{E2}^2(t)$ represents the costs of the cooperative country from transporting the black tiger shrimp. $lx_{M2}(t)$ represents the reputation of the cooperative country from controlling the black tiger shrimp in the "bringing to table" mode.

The change in the reputation of US and its cooperative country under the mode of "bringing to table" can be expressed as:

$$\dot{x}_{E1}(t) = f_1 G_{E1}(t) - \delta x_{E1}(t) \tag{11}$$

$$\dot{x}_{E2}(t) = (f_E + f_2) G_{E2}(t) - \delta x_{E2}(t) \tag{12}$$

In the above formula, $f_1 G_{E1}(t)$ represents the increase in the reputation of the US for tackling the black tiger shrimp crisis in the "bringing to table" mode. $\delta x_{E1}(t)$ represents the decline in the reputation of the US in the "bringing to table" mode. $f_E G_{E2}(t)$ represents the satisfaction of the residents of the partner country for eating the black tiger shrimp. $f_2 G_{M2}(t)$ represents the increase in the reputation of the partner country for tackling the black tiger shrimp crisis in the "bringing to table" mode. $\delta x_{E2}(t)$ represents the decline in the reputation of the partner country in the "bringing to table" mode.

## 3. Results

In the differential game, the control of the US and its cooperative country on black tiger shrimp is not only affected by control variables and parameters, but also changes over time. In order to better calculate the control quantity and social benefits, the HJB formula is used. The HJB formula is a partial differential equation, which is the core of optimal control.

### 3.1 HJB formula

Under the mode of introducing natural enemy, the HJB equation of the social welfare function of the US and cooperative country are:

$$\rho V_{B1} = \max_{G_{B1}(t)} \left\{ \left[ b_1 G_{B1}(t) - \frac{c_1}{2} G_{B1}^2(t) + lx_{B1}(t) \right] + \frac{\partial V_{B1}}{\partial x_{B1}} [f_1 G_{B1}(t) - \delta x_{B1}(t)] \right\} \tag{13}$$

$$\rho V_{B2} = \max_{G_{B2}(t)} \left\{ \left[ b_2 G_{B2}(t) + b_2 \ln(1 + h_B) G_{B2}(t) - \frac{c_2 + c_B}{2} G_{B2}^2(t) + lx_{B2}(t) \right] + \frac{\partial V_{B2}}{\partial x_{B2}} [(f_2 - f_B) G_{B2}(t) - \delta x_{B2}(t)] \right\} \tag{14}$$

Under the mode of making feed, the HJB equation of the social welfare function of the US and cooperative country are:

$$\rho V_{M1} = \max_{G_{M1}(t)} \left\{ \left[ b_1 G_{M1}(t) - \frac{(c_1 + c_{M1})}{2} G_{M1}^2(t) + q_{M1} G_{M1}(t) + l x_{M1}(t) \right] + \frac{\partial V_{M1}}{\partial x_{M1}} [f_1 G_{M1}(t) - \delta x_{M1}(t)] \right\} \quad (15)$$

$$\rho V_{M2} = \max_{G_{M2}(t)} \left\{ \left[ q_{M2} G_{M2}(t) - \frac{c_{M2}}{2} G_{M2}^2(t) + l x_{M2}(t) \right] + \frac{\partial V_{M2}}{\partial x_{M2}} [f_2 G_{M2}(t) - \delta x_{M2}(t)] \right\} \quad (16)$$

Under the mode of "bringing to table", the HJB equation of the social welfare function of the US and cooperative country are:

$$\rho V_{E1} = \max_{G_{E1}(t)} \left\{ \left[ b_1 G_{E1}(t) - \frac{c_1}{2} G_{E1}^2(t) + q_{E1} G_{E1}(t) + l x_{E1}(t) \right] + \frac{\partial V_{E1}}{\partial x_{E1}} [f_1 G_{E1}(t) - \delta x_{E1}(t)] \right\} \quad (17)$$

$$\rho V_{E2} = \max_{G_{E2}(t)} \left\{ \left[ q_{E2} G_{E2}(t) - \frac{c_T}{2} G_{E2}^2(t) + l x_{E2}(t) \right] + \frac{\partial V_{E2}}{\partial x_{E2}} [(f_E + f_2) G_{E2}(t) - \delta x_{E2}(t)] \right\} \quad (18)$$

## 3.2 Result of equilibrium

Proposition 1: Under the mode of introducing natural enemy, the control degree of black tiger shrimp and social benefits of US and cooperative country are respectively (the specific solving procedure is shown in S1 File):

$$G_{B1}^*(t) = \frac{b_1 + \frac{l}{\rho + \delta}}{c_1} \quad (19)$$

$$G_{B2}^*(t) = \frac{b_2 + b_2 \ln(1 + h_B) + \frac{l}{\rho + \delta}(f_2 - f_B)}{c_2 + c_B} \quad (20)$$

$$V_{B1}^* = \frac{l}{\rho + \delta} x_{B1} + \frac{1}{\rho} b_1 \frac{b_1 + \frac{l}{\rho + \delta}}{c_1} - \frac{c_1}{2} \frac{1}{\rho} \left( \frac{b_1 + \frac{l}{\rho + \delta}}{c_1} \right)^2 + \frac{l}{\rho + \delta} \frac{1}{\rho} f_1 \frac{b_1 + \frac{l}{\rho + \delta}}{c_1} \quad (21)$$

$$\begin{aligned}
V_{B2}^* &= \frac{1}{\rho} b_2 \frac{b_2 + b_2 \ln(1 + h_B) + \frac{l}{\rho + \delta}(f_2 - f_B)}{c_2 + c_B} + \frac{1}{\rho} b_2 \ln(1 + h_B) \frac{b_2 + b_2 \ln(1 + h_B) + \frac{l}{\rho + \delta}(f_2 - f_B)}{c_2 + c_B} \\
&\quad - \frac{c_2 + c_B}{2} \frac{1}{\rho} \left[ \frac{b_2 + b_2 \ln(1 + h_B) + \frac{l}{\rho + \delta}(f_2 - f_B)}{c_2 + c_B} \right]^2 + \frac{l}{\rho + \delta} x_{B2} \\
&\quad + \frac{l}{\rho + \delta} \frac{1}{\rho} (f_2 - f_B) \frac{b_2 + b_2 \ln(1 + h_B) + \frac{l}{\rho + \delta}(f_2 - f_B)}{c_2 + c_B}
\end{aligned} \quad (22)$$

Conclusion 1: The reproductive rate of natural enemies is proportional to the degree of control of black tiger shrimp in the cooperative country. The greater the negative impact of natural enemies reduction per unit on the ecology and reputation of the cooperative country, the lower the degree of control of black tiger shrimp in the cooperative country.

Proposition 2: Under the mode of making feed, the control degree of black tiger shrimp and social benefits of US and cooperative country are respectively (the specific solving procedure is shown in S2 File):

$$G^*_{M1}(t) = \frac{b_1 + q_{M1} + \frac{l}{\rho+\delta}f_1}{c_1 + c_{M1}} \tag{23}$$

$$G^*_{M2}(t) = \frac{q_{M2} + f_2\frac{l}{\rho+\delta}}{c_{M2}} \tag{24}$$

$$V^*_{M1} = \frac{l}{\rho+\delta}x_{M1} + \frac{1}{\rho}b_1\frac{b_1 + q_{M1} + \frac{l}{\rho+\delta}f_1}{c_1 + c_{M1}} - \frac{1}{\rho}\frac{(c_1 + c_{M1})}{2}\left(\frac{b_1 + q_{M1} + \frac{l}{\rho+\delta}f_1}{c_1 + c_{M1}}\right)^2$$

$$+ \frac{1}{\rho}q_{M1}\frac{b_1 + q_{M1} + \frac{l}{\rho+\delta}f_1}{c_1 + c_{M1}} + \frac{1}{\rho}\frac{l}{\rho+\delta}f_1\frac{b_1 + q_{M1} + \frac{l}{\rho+\delta}f_1}{c_1 + c_{M1}} \tag{25}$$

$$V^*_{M2} = \frac{l}{\rho+\delta}x_{M2} + \frac{1}{\rho}q_{M2}\frac{q_{M2} + f_2\frac{l}{\rho+\delta}}{c_{M2}} - \frac{c_{M2}}{2}\frac{1}{\rho}\left(\frac{q_{M2} + f_2\frac{l}{\rho+\delta}}{c_{M2}}\right)^2$$

$$+ \frac{l}{\rho+\delta}\frac{1}{\rho}f_2\frac{q_{M2} + f_2\frac{l}{\rho+\delta}}{c_{M2}} \tag{26}$$

Conclusion 2: The price of feed is in direct proportion to the control degree of black tiger shrimp. The cost of feed production is in inverse proportion to the control degree of black tiger shrimp.

Proposition 3: Under the mode of "bringing to the table", the control degree of black tiger shrimp and social benefits of US and cooperative country are respectively (the specific solving procedure is shown in S3 File):

$$G^*_{E1}(t) = \frac{b_1 + q_{E1} + \frac{l}{\rho+\delta}f_1}{c_1} \tag{27}$$

$$G^*_{E2}(t) = \frac{q_{E2} + \frac{l}{\rho+\delta}(f_E + f_2)}{c_T} \tag{28}$$

$$V^*_{E1} = \frac{l}{\rho+\delta}x_{E1} + \frac{1}{\rho}b_1\frac{b_1 + q_{E1} + \frac{l}{\rho+\delta}f_1}{c_1} - \frac{c_1}{2}\frac{1}{\rho}\left[\frac{b_1 + q_{E1} + \frac{l}{\rho+\delta}f_1}{c_1}\right]^2 + q_{E1}\frac{1}{\rho}\frac{b_1 + q_{E1} + \frac{l}{\rho+\delta}f_1}{c_1}$$

$$+ \frac{l}{\rho+\delta}\frac{1}{\rho}f_1\frac{b_1 + q_{E1} + \frac{l}{\rho+\delta}f_1}{c_1} \tag{29}$$

$$V_{E2}^* = \frac{l}{\rho+\delta}x_{E2} + \frac{1}{\rho}q_{E2}\frac{q_{E2}+\dfrac{l}{\rho+\delta}(f_E+f_2)}{c_T} - \frac{c_T}{2}\frac{1}{\rho}\left[\frac{q_{E2}+\dfrac{l}{\rho+\delta}(f_E+f_2)}{c_T}\right]^2$$

$$+\frac{l}{\rho+\delta}\frac{1}{\rho}(f_E+f_2)\frac{q_{E2}+\dfrac{l}{\rho+\delta}(f_E+f_2)}{c_T} \tag{30}$$

Conclusion 3: The higher the price of black tiger shrimp, the higher the level of control in the United States and its cooperative country. The higher the satisfaction of eating black tiger shrimp in the cooperative country, the higher the level of control of black tiger shrimp.

## 3.3 Numerical analysis

In order to describe the change of social utility of US and its cooperative country in more detail in the process of controlling the tiger shrimp invasion,, numerical analysis is used in this article. This article makes the following assumptions about the relevant parameters.

Differential games will be used to find the optimal strategy of each party by modeling and numerically calculating the utility function. There are several main benefits of performing numerical analysis in this process. First, specific optimization. Numerical analysis will use mathematical models and computers to assign specific values to make the results of the model more explicit, making it easier to find local or even global optimal solutions. This will help all parties to find the strategy that best serves their interests. Second, explicit comparison. Through the setting of the utility function and numerical analysis, the strategies of the parties and their results can be visualized in numerical form, which not only compares the advantages and disadvantages of different strategies, but also intuitively observes the impact of strategy adjustment on the change of utility value. Third, theoretical prediction. With numerical analysis, parties can also compare and improve strategies to optimize their interests and continuously improve the utility value, while being able to predict and study possible future scenarios. Fourth, resolving discrimination. Numerical analysis can quantify an individual's strategic choices and the advantages or losses they bring in the game process, which helps to find fair solutions.

Controlling black tiger shrimp populations often requires interventions in the ecosystem, which can involve costs for scientific research, environmental monitoring, harvesting equipment, labor, and many other aspects. These are high costs, especially when the black tiger shrimp has a wide distribution range, large numbers and complex biological characteristics, the difficulty and cost of control is more likely to rise. In addition to this, an environmental impact assessment may be required before control measures are implemented to avoid other ecological problems that may arise after the measures are implemented. For a species like the black tiger shrimp, which occupies an important role in the ecosystem, any manipulation measures need to be treated with caution, which increases the overall control costs. After the control measures are implemented, follow-up management and monitoring is also required, such as regular assessment of the control effects and timely adjustment of the control strategy. And this will be a considerable cost in the long run. Meanwhile, the cost of producing a unit quantity of feed is relatively low, mainly because of the following reasons. First, economies of scale. Feed production is a large-scale, industrialized process that can reduce unit costs through economies of scale [26]. Second, technological progress. Modern feed production technology is becoming more and more mature, which can effectively reduce production costs. Third, raw

material access. Compared with the control organisms, the raw materials of feed are more abundant and easy to obtain, which also further reduces the production cost. In summary, the cost of controlling the number of black tiger shrimp is greater than the cost of producing a unit quantity of feed. Therefore, this paper assumes that the cost $c_1, c_2$ of the US or the cooperative country from the control of unit quantity of black tiger shrimp is 2.5; the cost $c_{M1}, c_{M2}$ of the US or the cooperative country to produce the feed per unit quantity is 2. Some possible reasons for the higher transportation costs of black tiger shrimp include. First, black tiger shrimp is a live product, need to maintain a certain living environment, such as suitable temperature, humidity, etc., the transportation process requires good insulation and ventilation facilities, compared with the transportation of ordinary goods, equipment and average operating costs are higher. Secondly, because the transportation process needs to ensure the survival rate of shrimp, transportation time requirements are very strict, often need to choose a faster but also more expensive mode of transportation, such as air transport. Third, the packaging of black tiger shrimp needs to meet the requirements of its survival environment while ensuring transportation safety, and the cost of packaging tends to be higher compared to ordinary products [27]. As a result, these special needs and constraints make the transportation of black tiger shrimp more expensive. Therefore, this paper assumes that the transportation cost $c_T$ of the black tiger shrimp per unit quantity is 2.5.

For the vast majority of people, the satisfaction and gratification that comes from consuming black tiger prawns is immediately apparent. People love and appreciate food intuitively and this satisfaction can be experienced through direct sensory experience [28]. Black tiger prawns have gained worldwide popularity due to their delicious taste and nutritional value. This popularity may outweigh its possible ecological impacts. Many people may not be aware of the negative impacts that a reduction in the number of natural enemies of the black tiger shrimp may have on the reputation of cooperating countries. Community, national, and geographical ecological issues are often less easily understood and valued by the general public than intuitive food satisfaction because of their complexity and long-term nature. Therefore, this paper assumes that the satisfaction $f_E$ of the cooperative country from eating the black tiger shrimp per unit quantity is 1.5; the negative impact $f_B$ of the reduction of natural enemies per unit quantity on the reputation of the cooperative country is 1.

The other parameters are not the focus of this paper, while their magnitude will not affect the results of the analysis, thus, the paper makes the following assumptions. The revenue $b_1, b_2$ of the US or the cooperative country from the control of unit quantity of black tiger shrimp is 5. The discount rate $\rho$ that occurs over time is 0.9. Decay $\delta$ of reputation is 0.1. The positive impact $l$ of reputation per unit quantity is 1. The negative impact $c_B$ of the reduction of natural enemies per unit quantity on the ecology of the cooperative country is 0.5. The reproduction speed $h_B$ of the natural enemies is 2. The selling price $q_{M1}, q_{M2}$ of the US or the cooperative country to sell the feed is 3.

If the price $q_{E1}$ of black tiger shrimp sold directly in the US is 1, this article can calculate the social benefits of US:

$$V_{B1}^* = 6.33 + 2.67f_1 \tag{31}$$

$$V_{M1}^* = 0.12f_1^2 + 1.98f_1 + 9.16 \tag{32}$$

$$V_{E1}^* = 1 + 0.22(6+f_1)^2 \tag{33}$$

The following Fig 2 can also be produced:

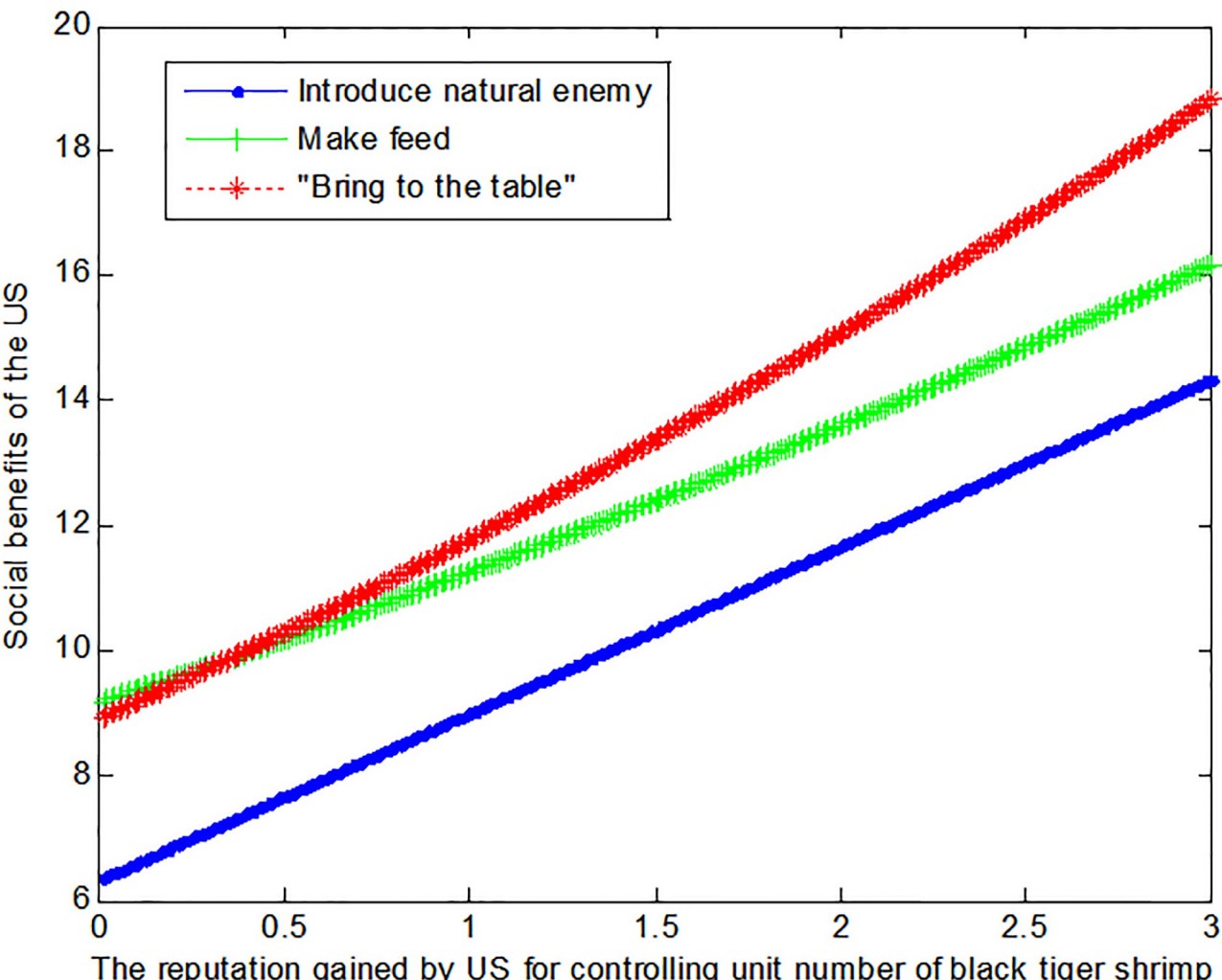

**Fig 2. Impact of reputation on the social benefits of US.**

If the price $q_{E1}$ of black tiger shrimp sold directly in the US is 4, this article can calculate the social benefits of US:

$$V_{B1}^* = 6.33 + 2.67f_1 \tag{34}$$

$$V_{M1}^* = 0.12f_1^2 + 1.98f_1 + 9.16 \tag{35}$$

$$V_{E1}^* = 1 + 0.22(9+f_1)^2 \tag{36}$$

The following Fig 3 can also be produced:

Conclusion 4: The US can benefit more from the mode of making feed if the price of the black tiger shrimp and the reputation of US regulation of black tiger shrimp are lower. Otherwise, the US prefers to sell the black tiger shrimp directly and thus "bring to the table".

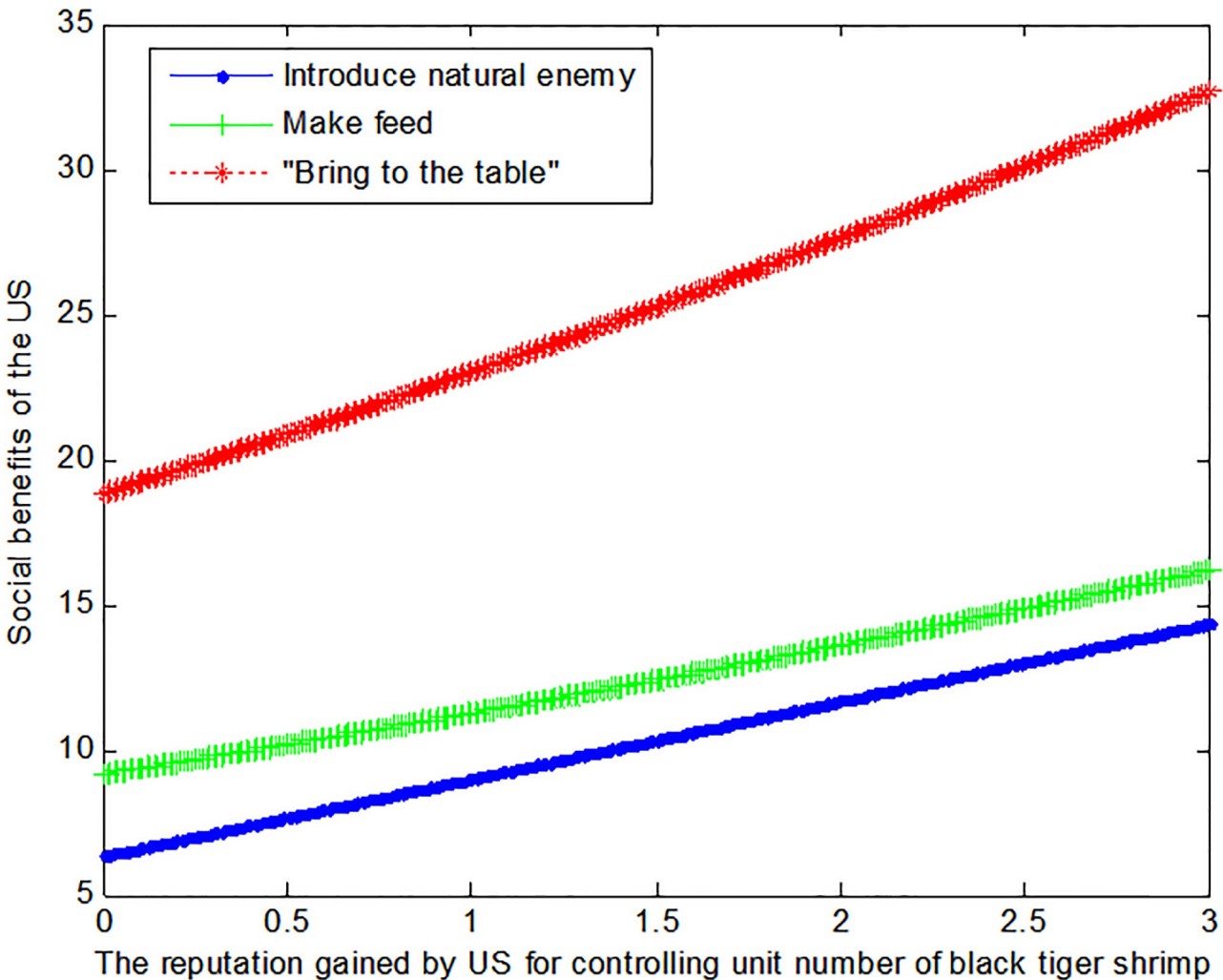

**Fig 3. Impact of reputation on the social benefits of US.**

If the price $q_{E2}$ of black tiger shrimp sold directly in the cooperative country is 1, this article can calculate the social benefits of the cooperative country:

$$V_{B2}^* = 1 + 0.185(f_2 + 9.5)^2 \tag{37}$$

$$V_{M2}^* = 1 + 0.28(f_2 + 3)^2 \tag{38}$$

$$V_{E2}^* = 1 + 1.39(1 + 0.4f_2)^2 \tag{39}$$

The following Fig 4 can also be produced:

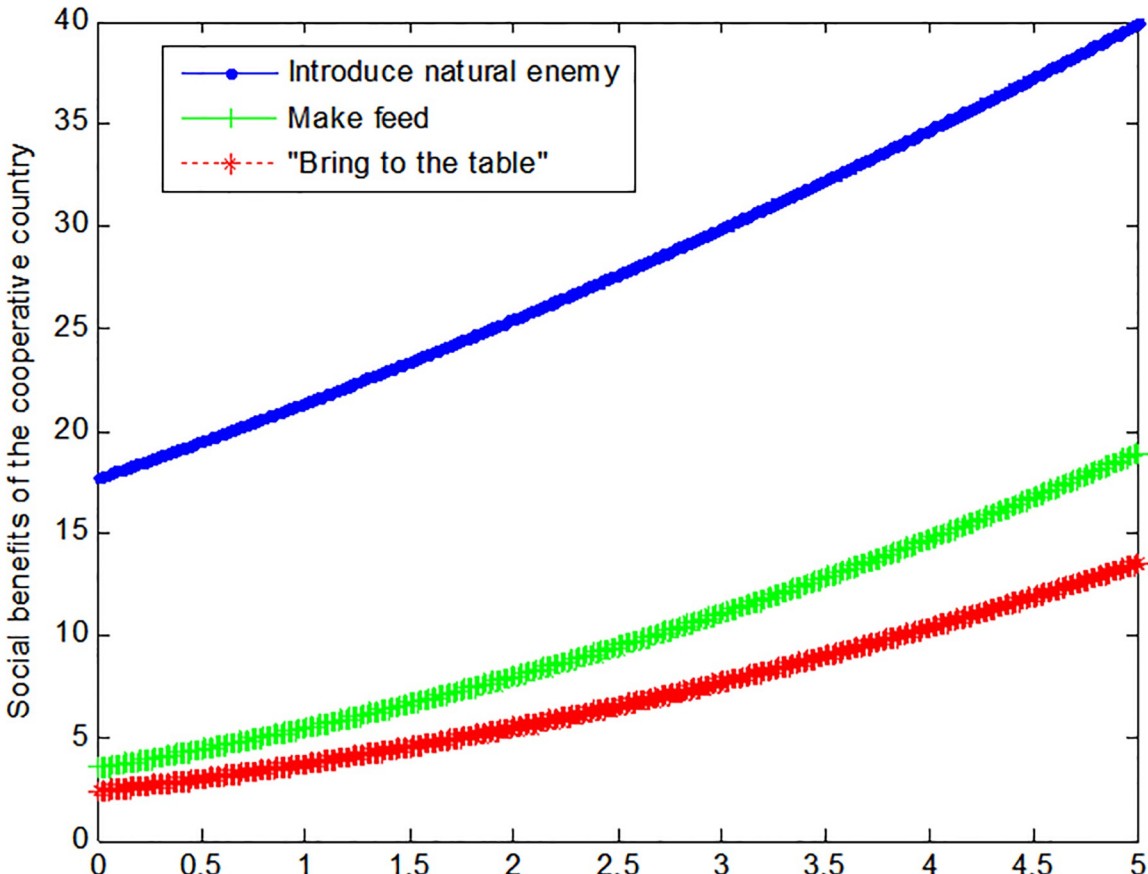

**Fig 4. Impact of reputation on the social benefits of the cooperative country.**

If the price $q_{E2}$ of black tiger shrimp sold directly in the cooperative country is 4, this article can calculate the social benefits of the cooperative country:

$$V_{B2}^* = 1 + 0.185(f_2 + 9.5)^2 \tag{40}$$

$$V_{M2}^* = 1 + 0.28(f_2 + 3)^2 \tag{41}$$

$$V_{E2}^* = 1 + 1.39(2.2 + 0.4f_2)^2 \tag{42}$$

The following Fig 5 can also be produced:

Conclusion 5: Rather than the mode of making feed or "bring to the table", the cooperative country hope to control the tiger shrimp invasion by introducing natural enemy.

## 4. Discussion

When the reproductive rate of natural enemies of black tiger shrimp is higher, the cooperative country should increase the introduction of the natural enemies. The above content is the main viewpoint of Conclusion 1. This is somewhat different from the research results of Yuan et al. [29]. Yuan et al. [29] believed that the introduction of natural enemies should depend on

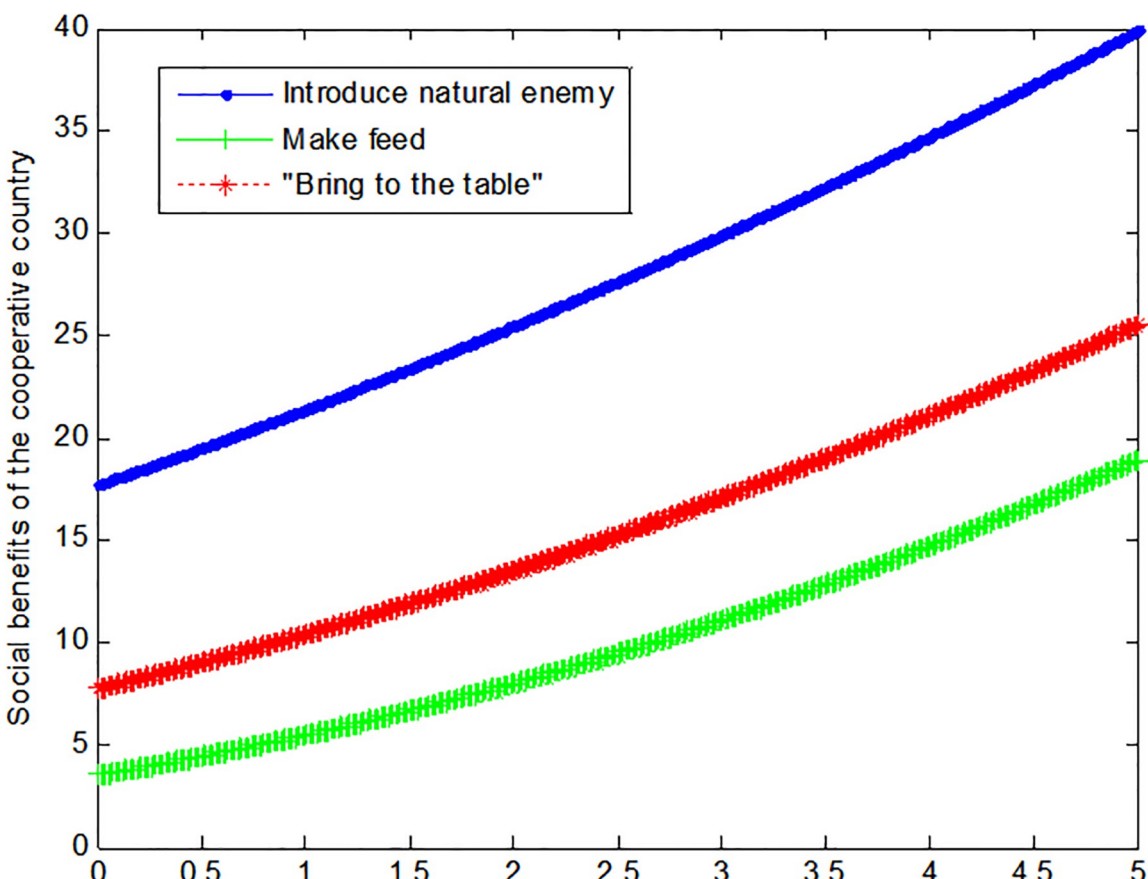

**Fig 5. Impact of reputation on the social benefits of the cooperative country.**

the environment, rather than the reproductive rate mentioned in this article. This is mainly because the research objects of the two are different, and the environments of natural enemies are not consistent. Yuan et al. [29] studied the abundance and impacts of four adenosine sites in Southwest China, which were invaded at different times and supported different densities of insects, and produced corresponding data. The study in this paper focuses on mathematical modeling to construct a strategy space and derive how to manage the black tiger shrimp infestation. Specifically, this can be explained by the following main reasons. First, improve the control effect. Natural enemies are a natural biological control method, which take black tiger shrimp as food and can effectively control the population of black tiger shrimp. When the reproductive rate of natural enemies increases, the introduction of more natural enemies can improve the control effect of black tiger shrimp, because the increase in the number of natural enemies means that more individuals can prey on black tiger shrimp, thus more effectively controlling its population. Second, balance the ecosystem. As an invasive species, black tiger shrimp may destroy the balance of the local ecosystem and pose a threat to other organisms [8]. By introducing more natural enemies, the balance of the ecosystem can be restored and maintained. The increase in natural enemies will increase the predation pressure on black tiger shrimp, which helps to restore the diversity and ecological balance of local species. The ecosystem can be balanced by selecting suitable natural enemies. For example, to manage pests

in tea gardens, for example, the dominant natural enemies of the same pest in different varieties of tea gardens in the same region are often different, and it is necessary to select suitable natural enemies to effectively manage the ecosystem of tea gardens [30]. Third, strengthen the prevention and control strategy. The cooperative countries recognize the importance of natural enemies in the management of the black tiger shrimp and that increasing the number of natural enemies can increase the control of the black tiger shrimp. Therefore, the cooperative country may increase investment and take more measures to introduce and cultivate suitable natural enemies, such as increasing the number of natural enemies through artificial breeding and releasing. For example, the extensive use of the natural enemy mesquite to inoculate Carpobrotus edulis can play a very important role in the control of Carpobrotus edulis [3]. Fourth, research and monitoring. The cooperative country may also strengthen the research and monitoring of natural enemies to understand the reproduction and ecological characteristics of natural enemies, so as to better adjust the intensity and strategy of introducing natural enemies. For example, control strategies for crop pests are influenced by the frequency of pest outbreaks, ET, and the intensity of environmental disturbances [31]. Through scientific research and monitoring, the cooperative country can more accurately judge the interaction between natural enemies and black tiger shrimp, and take corresponding control measures. In a word, when the reproduction rate of natural enemies of black tiger shrimp is higher, the cooperative country realize that the introduction of more natural enemies can improve the control effect of black tiger shrimp, and take corresponding measures to increase the number of natural enemies. This can more effectively control the population number of black tiger shrimp, and protect the stability and diversity of the local ecosystem at the same time.

In the case of tiger shrimp overpopulation, the higher the price of feed made from tiger shrimp, the greater the degree of fishing for tiger shrimp may be. The above content is the main point of conclusion 2. This is different from the study of Bowzer et al. [17], who believed that the nutritional composition and shelf stability of Asian carp as raw materials must be determined. These differences are mainly because when tiger shrimp are in excessive numbers, they may have a negative impact on the local ecosystem, including damage to other aquatic resources and destruction of ecological balance. At this time, feed made from tiger shrimp may be a solution to control their numbers by fishing for tiger shrimp. Due to high demand, the price of feed may rise, thereby stimulating fishing for tiger shrimp to meet the demand of society for feed. Meanwhile, feed made from tiger shrimp has a wide range of uses in the agricultural and aquaculture industries. First, aquaculture. As a high-quality protein source, tiger shrimp is often used as a feed in aquaculture. Tiger shrimp feed can provide rich nutrition, helping the growth and development of aquaculture products. They can be used for the cultivation of fish, shrimp, shellfish, etc., to improve their growth rate and meat quality. Second, poultry farming. Black tiger shrimp feed can also be used for poultry farming, such as raising chickens, ducks and so on. They contain high protein and essential amino acids, which can provide rich nutrition for poultry and promote their growth and meat quality. Third, pet food. Black tiger shrimp feed can also be used to make pet food, such as dog food, cat food and so on. They provide a balanced nutritional combination to meet the needs of pets and contribute to the health and growth of pets. Fourth, organic fertilizer. In the field of agriculture, black tiger shrimp feed is also used to make organic fertilizer. They are rich in organic matter and trace elements, which can be used as high-quality organic fertilizer to provide the nutrition needed for plant growth. In summary, the feed made with black tiger shrimp has wide applications in agriculture and aquaculture, which can provide rich nutrition, promote growth and improve meat quality. Therefore, in the case of black tiger shrimp flooding, the higher the price of the feed made with black tiger shrimp, the greater the degree of fishing for black tiger shrimp may be. However, aspects such as market dynamics and consumer behavior also need

to be taken into account regarding the production of black tiger shrimp feed. Firstly, market dynamics are analyzed in terms of market dynamics. Firstly, the demand for seafood, including black tiger prawns, has been steadily increasing in line with global population growth and increased demand for protein intake. This means that the market demand for black tiger shrimp feed continues to grow [32]. Secondly, market demand for more efficient and nutritionally balanced feeds has driven innovations in feed formulations, such as the incorporation of more versatile ingredients such as shrimp meal. Thirdly, the growing concern of consumers and environmental organizations about the sustainability of aquaculture has led to the emergence of more feeds produced from sustainable ingredients, such as alternative protein sources instead of traditional fishmeal and fish oil. Also, the use of black tiger prawns as feed has to be considered for sustainability. Then in consumer behavior is analyzed. First, more and more consumers are concerned about the sustainability of their products, which leads the aquaculture industry to pursue more environmentally friendly feed sources, especially in markets with high demands on environmental and social responsibility. Second, consumers are increasingly concerned about food safety and nutritional value, which encourages feed manufacturers to develop high-quality feeds that improve shrimp quality and health status. Third, economic pressures and consumer price sensitivity can influence demand for higher-priced feed products, especially in highly competitive markets.

If the black tiger shrimp is abundant, the cooperative country will increase the degree of fishing for black tiger shrimp if they are more satisfied with eating black tiger shrimp. The above is the main idea of conclusion 3. This is inconsistent with the view of Sani et al. [33]. Sani et al. [33] believed that shrimp consumption is not only limited by taste, but also affected by microbial risks. However, this view is held in this article for the following reasons. When the black tiger shrimp is in excess, they may have a negative impact on the local ecosystem, including damage to other aquatic resources and destruction of ecological balance [20]. In this case, the cooperative country may encourage the fishing of black tiger shrimp to control its quantity and reduce its impact on the ecosystem. If the demand for black tiger shrimp for eating increases in the cooperative country, fishermen may actively participate in the fishing of black tiger shrimp to meet the market demand. This may include increasing the fishing force, investing more resources and equipment to catch black tiger shrimp. However, it should be noted that over-fishing of black tiger shrimp may cause further damage to its population quantity and ecosystem. Therefore, when increasing the degree of fishing, the cooperative country should take reasonable management and protection measures to ensure the sustainability of fishing activities and prevent overuse of resources and further damage to the ecosystem. However, this approach to address the black tiger prawn flooding through harvesting may pose potential sustainability issues and require an appropriate regulatory framework to ensure the sustainability and effectiveness of the harvesting behavior. The following are some of the potential sustainability detail issues. First, overfishing may lead to further negative impacts on biodiversity, for example, fishing practices may destroy habitat or inadvertently catch other non-target species. Second, in the process of fishing for black tiger prawns, there is the potential for catching other marine organisms, which may have negative impacts on non-target species. Third, if fishing becomes profitable, it may induce overfishing and exacerbate ecological problems rather than address the problem of invasive species. Fourth, the introduction of fishing as a means of control may reduce black tiger shrimp populations in the short term, but may lead to resource depletion or other environmental problems in the long term. The following is the regulatory framework for managing fishing behavior. First, fishing quotas are established to control the total amount of fishing and to ensure that fishing activities do not cause irreversible damage to the ecosystem. Second, specific fishing seasons and areas are established to protect important breeding and rearing areas, as well as to avoid fishing activities in other

sensitive ecological zones. Third, strict control over fishing gears and methods, such as banning the use of bottom trawls, which are more environmentally destructive. Fourth, full monitoring of fishing behavior and real-time data collection should be realized to ensure that all activities are accurately recorded and managed [34]. The above components of the regulatory framework are not exhaustive and should be specifically designed and adapted to the specific circumstances and ecological habits of the species targeted by the fishery. Importantly, any fishing methods and management measures should be based on ecological principles and best sustainable practices to ensure that the problem of invasive species is not only addressed, but that the integrity and functioning of marine ecosystems are protected.

According to conclusion 4, if the selling price of black tiger shrimp and the reputation of the US for controlling black tiger shrimp are low, then the US can gain more benefits under the mode of making feed. Otherwise, the US prefers to sell black tiger shrimp directly, thus putting them directly "bring to the table". This is inconsistent with the viewpoint of Islam et al. [35]. Islam et al. [35] believed that heavy metal pollution has an impact on the sale of black tiger shrimp, while this article does not address the problem of pollution of black tiger shrimp. Of course, heavy metal contamination has several potential impacts on black tiger shrimp. First, consumer health concerns. If consumers learn that black tiger shrimp are contaminated with heavy metals, this may cause them to worry about the possible adverse health effects of consuming contaminated shrimp. As a result, they may reduce or stop purchasing black tiger shrimp [18]. Second, sales decline. As more consumers avoid potentially contaminated products, overall sales of black tiger shrimp may decline. Third, price volatility. Reduced demand may cause the market price of black tiger shrimp to fall. However, if suppliers reduce the number of contaminated black tiger prawns on the market to maintain product safety, this could lead to an increase in price due to shortages. Fourth, brand and reputation loss. If a brand is found to have high levels of heavy metal contamination in its black tiger shrimp products, this could damage the company's reputation and lead to a reduction in long-term sales. It is therefore necessary for farmers, suppliers, retailers and interested parties to ensure that seafood produced and sold meets food safety standards in order to maintain market stability and consumer trust. To determine whether to sell black tiger shrimp directly or make them into feed, several factors need to be considered. First, market demand. First of all, this requires studying the market demand for black tiger shrimp and feed. If there is a high market demand for black tiger shrimp, selling them may bring better benefits. In contrast, if there is a greater market demand for black tiger shrimp feed, making it into feed may be more profitable. Second, cost-effectiveness. This also requires comparing the cost-effectiveness between directly selling black tiger shrimp and making them into feed. Factors such as production cost, transportation cost and market price are considered to calculate the cost and benefits of selling black tiger shrimp and making feed. Third, environmental impact. The spread of black tiger shrimp may have certain impacts on the ecosystem. Understand how much impact the shrimp have on the local ecological environment and whether they are considered an invasive species. If the shrimp pose a serious threat to the local ecological environment, making them into feed may be a more sustainable solution. Fourth, feasibility. This requires consideration of the feasibility of making black tiger shrimp into feed and the availability of relevant technology and equipment. If making black tiger shrimp into feed requires a lot of resources, technology and equipment, and the corresponding ability or resources are not available, then selling them directly may be a simpler and more feasible option. Taking the above factors into consideration, a more informed decision can be made on whether to sell black tiger shrimp or make them into feed.

When studying the black tiger shrimp problem, if researchers only consider situations where natural enemy introductions do not negatively impact local ecology in the America, and

ignore feed logistics costs, this could lead to the following. First, incompleteness of analysis. Failure to consider the full range of relevant environmental and economic influences, such as feed logistics costs, can lead to incomplete findings. Second, bias in risk estimation. Risk assessment by eradicating the local ecology may be subject to, but ignoring the costs or potential economic impacts of feed logistics can result in biased risk assessment. Third, applicability limitations. The study conclusions may only be applicable to biocontrol interventions that operate harmlessly under specific conditions and need to be applied with caution in practice, and by ignoring logistics costs, the conclusions may not be applicable to contexts where cost constraints are a major issue [36]. Fourth, potentially uneconomic options. If feed logistics costs are significant, then biological control may not be economically feasible even if it is ecologically safe. From there, these limitations will be taken into account in future studies.

The results between the control strategies of the equilibrium results and their comparison can lead to relevant strategies for the relevant sectors. First, if the market price of black tiger prawns is low, then the returns from selling black tiger prawns directly as a food product may also be low, and therefore converting black tiger prawns into feed would bring more economic benefits, especially if there is demand from the feed industry. Secondly, the mode of making feed may require a large upfront investment, including the establishment of a collection system, treatment facilities and a distribution network [37]. However, once established, returns are likely to be more stable and long-term. Selling shrimp directly as a food product, while potentially more profitable, would require more resources to be invested in marketing and product development. Third, if the America has a low regulatory reputation for black tiger shrimp, direct sales of black tiger shrimp could raise consumer concerns about the safety and quality of the product, which would affect sales. In contrast, converting black tiger shrimp to feed may have a lesser impact on public image. Fourth, the cooperating countries' desire to introduce natural enemies to control the black tiger shrimp population is motivated by environmental protection or maintaining ecological balance. If the America disagrees with the cooperating country on the strategy, it needs to coordinate through communication to find a solution that can benefit both parties. In summary, based on the information provided, the America may prefer the model of converting black tiger shrimp into feed to gain more economic benefits and protect the company's reputation. However, the relationship with and expectations of cooperating countries also need to be taken into account, complicating the strategic choice. Through in-depth analysis and discussion, it is possible to find an approach that takes all factors into account and achieves economic benefits while preserving the cooperative relationship and protecting the environment at the same time.

Considering that the cooperating countries prefer to control the black tiger shrimp infestation through the natural enemy introduction model, the following are some possible specific policy recommendations. First, invest in scientific research to study the habits, growth rate and reproductive capacity of the black tiger shrimp, as well as to study its possible natural enemies, in order to determine the most suitable method to control its population. Secondly, a stable monitoring system should be set up to report changes in the black tiger shrimp population in a timely manner so that natural enemies can be introduced as needed. At the same time, the monitoring of the population of natural enemies that may have a negative impact on the environment should be strengthened. Thirdly, education and training should be provided to relevant departments and decision makers involved in the control of black tiger shrimp so that they can fully understand the reproduction pattern of black tiger shrimp and the possible impacts caused by the introduction of natural enemies. Fourth, conduct extensive public outreach to enhance public understanding of the black tiger shrimp infestation problem and encourage their participation in the protection of the local ecosystem, for example, through the establishment of special fishing seasons and the reduction of black tiger shrimp

overfishing. Fiffthly, relevant legislation should be introduced to regulate the process of natural enemy introduction to prevent new ecological problems that may result, such as the invasion of non-native species. Similarly, penalties should be imposed on illegal fishing and sale of wild black tiger prawns to support efforts to control the population [38]. Sixth, cross-sectoral and cross-regional cooperation should be realized to share resources and information in order to control the black tiger prawn population more effectively. In addition, there may be a need to share best practices and success stories with international partners. Such strategies can help manage black tiger shrimp populations while minimizing environmental impacts.

## 5. Conclusion

This article hypothesizes that the problem of black tiger shrimp invasion can be controlled through three modes: introducing natural enemies, making feed and "bringing to the table". Considering the continuous reproduction of black tiger shrimp and the continuous changes of decisions of the United States and the cooperative country, this article constructs a differential game mode of the three modes: introducing natural enemies, making feed and "bringing to the table". The results show that the higher the market price of black tiger prawns, the more stringent the industry's regulatory and control measures for their production and distribution are to ensure that higher profits can be realized from the high price; if the price of black tiger shrimp is low or reputationally risky, the United States is likely to be more inclined to produce feed to minimize risk and achieve stable returns; when the price of black tiger shrimp rises and the reputation of the shrimp is good, it may be more profitable to sell it directly as a food product; biological control is often a more sustainable form of management that reduces damage to the environment.

The research in this article can also be generalized. For example, this article only considers the cases where the introduction of natural enemies will not cause negative impacts on the local ecology in the United States and the cost of feed logistics is ignored. In future research, the introduction of natural enemies will bring new ecological problems and the cost of feed logistics is higher, and related problems can be analyzed. Meanwhile, some blanks in the research can also be solved in future research. For example, the America acted first and the cooperating countries acted later, rather than simultaneously, to analyze how they managed the black tiger shrimp infestation. Several policy recommendations can also be made based on the conclusions, aimed at balancing economic benefits with ecological impacts. First, a flexible market regulation policy. Policies should be formulated to take into account price fluctuations of black tiger shrimp, feed prices, and the international reputation of the United States for controlling black tiger shrimp. Encourage direct sales when black tiger shrimp prices are high and reputation is good; promote utilization of black tiger shrimp as feed when prices are low or reputation is poor. Second, ecological risk assessment and management system. Establish a rigorous ecological risk assessment system that targets the introduction of natural enemies to control the black tiger shrimp to ensure that there is no negative impact on the local ecosystem. Third, logistics and supply chain optimization. Develop strategies to optimize feed logistics and supply chain to reduce costs. This will help maintain market competitiveness when using black tiger shrimp as feed and reduce potential negative impacts on the environment. Fourth, international cooperation intensification. To improve food safety and ecological security, intensify international cooperation in controlling black tiger shrimp. This can be achieved through shared research, technology exchange, common regulatory frameworks and trade agreements.

## Supporting information

**S1 File.**
(DOCX)

**S2 File.**
(DOCX)

**S3 File.**
(DOCX)

## Author Contributions

**Conceptualization:** Yuntao Bai.

**Data curation:** Yuntao Bai, Lan Wang.

**Formal analysis:** Yuntao Bai, Lan Wang, Delong Li.

**Funding acquisition:** Lan Wang, Delong Li.

**Investigation:** Yuntao Bai, Ruidi Hu, Lan Wang, Delong Li.

**Methodology:** Yuntao Bai, Ruidi Hu.

**Writing – original draft:** Yuntao Bai, Ruidi Hu.

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
