## [Decision Letter · Decision Letter 0]

22 Jan 2024

PONE-D-23-38997Analysis on the control of the black tiger shrimp in the America from the perspective of international cooperationPLOS ONE

Dear Dr. Wang,

Thank you for submitting your manuscript to PLOS ONE. After careful consideration, we feel that it has merit but does not fully meet PLOS ONE’s publication criteria as it currently stands. Therefore, we invite you to submit a revised version of the manuscript that addresses the points raised during the review process.

We look forward to receiving your revised manuscript.

Kind regards,

Mahmoud A.O. Dawood, PhD

Academic Editor

PLOS ONE

Journal Requirements:

"This work is financially supported by Social Science Planning Foundation of Chongqing in China (2021BS080); This work is financially supported by National Natural Science Foundation of China (72304157)."

Reviewers' comments:

Reviewer's Responses to Questions

**Comments to the Author**

1. Is the manuscript technically sound, and do the data support the conclusions?

Reviewer #1: Partly

Reviewer #2: Partly

2. Has the statistical analysis been performed appropriately and rigorously? 

Reviewer #1: N/A

Reviewer #2: Yes

3. Have the authors made all data underlying the findings in their manuscript fully available?

Reviewer #1: No

Reviewer #2: Yes

4. Is the manuscript presented in an intelligible fashion and written in standard English?

Reviewer #1: No

Reviewer #2: No

5. Review Comments to the Author

Reviewer #1: The manuscript investigates strategies for controlling the invasive black tiger shrimp through international cooperation between the America and other partner countries. The topic addresses an important issue and the goal of exploring cooperative approaches is meaningful. However, there are some substantial issues that need to be addressed for the manuscript to be suitable for publication.

Conceptual framework - While multiple control strategies are proposed, the theoretical/conceptual framework that guides comparison and analysis of the strategies is not clearly established. A stronger conceptual underpinning is needed to systematically examine the different approaches.

Methodology - The methodology section does not provide sufficient detail on the differential game models constructed. Key elements like model structure, assumptions, parameters, solution concepts etc. are missing. More robust and clearly described methodology is essential.

Literature review - Literature cited does not sufficiently synthesize previous findings on control of black tiger shrimp specifically. More focused review of related work is needed to position the current study appropriately. Also update the recent works in this field, such as:

[]Das, D. ., & Samanta, G. C. . (2024). An EOQ Model for Two Warehouse System During Lock-Down Considering Linear Time Dependent Demand. Transactions on Quantitative Finance and Beyond, 1(1), 15–28.

[]Aguirre-Pabón, J., Chasqui, L., Muñoz, E., & Narváez-Barandica, J. (2023). Multiple origins define the genetic structure of tiger shrimp Penaeus monodon in the colombian Caribbean Sea. Heliyon, 9(7).

[]Yun Tseng, H. ., Salam, N. . A. . ., Kazemi, M. ., Nadiheidari, H. ., & Mehlabani, E. G. . (2024). Reevaluating the Impact of Corporate Governance Mechanisms on Investment Management, Financing Strategies, and Corporate Performance. Management Analytics and Social Insights, 1(1), 1–16.

[]Vu, N. T., Jerry, D. R., Edmunds, R. C., Jones, D. B., & Zenger, K. R. (2023). Development of a global SNP resource for diversity, provenance, and parentage analyses on the Indo-Pacific giant black tiger shrimp (Penaeus monodon). Aquaculture, 563, 738890.

Analysis - Limited analysis is conducted to derive insights from the game models. Results and comparisons between the control strategies based on equilibrium outcomes are not clearly presented. Deeper strategic analysis is required.

Writing quality - Overall writing standard needs improvement in terms of clarity, flow, proper referencing etc. Manuscript contains repetitive information and lacks cohesion.

Policy relevance - While international cooperation is highlighted, practical policy dimensions and recommendations for partner countries are not sufficiently discussed.

Reviewer #2: The study presents an analysis of controlling the invasive black tiger shrimp in America, and as such, the article constructs three differential game models to analyze control strategies.

There are several critical points to consider:

Lines 31-33: There is some confusion here, are you referring to black tiger shrimp or Louisiana red-clawed shrimp? These are different species. The black tiger shrimp, scientifically known as Penaeus monodon, is native to Indo-Pacific, Asian, and Australian waters, not to the United States as mentioned in your introduction. This species is primarily marine, rather than freshwater, which contradicts the description in the article. Therefore, please add more sources to confirm the adaptations of this species to freshwater, or give salinity ranges based on past studies.

Also, while the introduction correctly identifies the black tiger shrimp as an invasive species in the United States, further emphasis on its marine nature and the specific impacts it has on the local ecosystems, particularly in southeastern U.S. waters, could strengthen the article.

The introduction discusses various control methods, including introducing natural predators, using the shrimp as feed, and promoting them as a food source. While these are valid points, it would be beneficial to incorporate a more detailed discussion on the FEASIBILITY and ecological consequences of these methods, especially the introduction of non-native predators, which can have unpredictable impacts on local ecosystems.

Lines 153-415: The section is quite extensive which is good for replicability and understanding the approach, but it might benefit from concise summarization. Are some of the protocols already done elsewhere and could be referenced?

Lines 155-177: The problem description is clear when you stated the significance of controlling black tiger shrimp, but the text could be more concise and focused on directly relating the problem to the methodology used.

Lines 264-285: The hypotheses listed are critical to the study but require a more robust justification. For instance, the assumption that introduced natural enemies will not negatively impact local ecology (Line 265) is significant and needs substantial evidence or references to similar studies.

Lines 318-321: The variables are well-defined, but.. it would be beneficial to explain why these specific variables were chosen and how they relate to the real-world scenario of black tiger shrimp control.

Lines 322-415: The differential game model is complex and seems appropriate for the study's aims. However, there's a lack of explanation about the choice of this model over other potential models and perhaps a better explanation of the formulations.

Lines 201-213, 364-367: The assessment of ecological impact, especially regarding the introduction of natural enemies, is mentioned but lacks depth, please try to reference to ecological impact studies.

Lines 214-235, 236-258: While the economic aspects are considered, the ecological feasibility and long-term sustainability of the proposed solutions (feed mode and 'bring to the table' mode) are not adequately addressed.

Lines 356-367, 382-391, 405-415: The use of reputation as a metric is interesting, but its operationalization and measurement in the context of ecological control are not sufficiently explained.

Lines 303-316: Ignoring feed logistics costs is a significant assumption that could affect the model's applicability in real-world scenarios. This needs justification or a discussion on potential impacts.

I recommend clarifying or explaining how abstract concepts like "reputation" are measured and factored into the models.

Lines 471-481: While you provide a list of parameters and their values, there is a lack of explanation regarding how these values were determined. Are they based on empirical data, expert estimation, or are they hypothetical? This needs to be clarified for the reader to understand the basis of your models.

Lines 467-520: This part of the section attempts to provide a more detailed description of social utility changes, but the transition from theoretical models to numerical analysis seems rough, perhaps a clearer explanation of how these numerical values are linked to the earlier models would be helpful?

Lines 487-514: This is optional but there seems to be a slight repetition of the formula for calculating social benefits under different prices for black tiger shrimp. While it's important to show these variations, the presentation could be more concise to avoid redundancy, to avoid confusing the readers.

Lines 522-527: The discussion acknowledges the difference in findings compared to Yuan et al. (2021) but please provide a better explanation for why these differences exist beyond stating the variance in research objects and environments.

Lines 528-547: The authors provide reasons for introducing natural enemies, but the discussion seems somewhat speculative. The arguments could be strengthened by including empirical evidence or case studies that support these strategies.

Lines 534-538: The impact on ecosystem balance is mentioned, but the discussion lacks specific examples or data showing how the introduction of natural enemies has historically affected ecosystems. This leaves the argument somewhat unsubstantiated.

Lines 554-578: The economic analysis of feed production from black tiger shrimp is interesting but lacks critical evaluation of market dynamics and consumer behavior. More rigorous economic analysis would add credibility to these claims.

Lines 579-594: The discussion on increasing fishing due to taste preferences (Lines 580-589) is overly simplistic and does not account for potential sustainability issues or regulatory frameworks governing fishing practices.

Lines 595-617: The comparison with the viewpoint of Islam et al. (2017) is noted, but the discussion does not adequately address the issue of heavy metal pollution in black tiger shrimp. This omission might undermine the reliability of the conclusions drawn about the profitability of different shrimp utilization methods.

Lines 619-630: The conclusion effectively summarizes the main findings regarding the control of black tiger shrimp, but it repeats much of what is already stated in the discussion without adding significant new insights or implications.

Lines 631-636: The authors acknowledge limitations in their research, such as not considering the negative impacts of introducing natural enemies and ignoring logistics costs but this section would benefit from a more direct discussion of how these limitations could affect the study's applicability or conclusions.

Lines 637-640: The suggestions for future research are relevant, but they are quite broad. More specific recommendations or potential research questions would be more helpful for guiding future studies in this area.

Lines 637-639: The conclusion mentions translating findings into policy recommendations but does not provide concrete suggestions or frameworks. The article could be strengthened by directly proposing policy changes or interventions based on the findings.

The work could be improved by providing more empirical evidence, addressing counterarguments more thoroughly, and offering more detailed but concise policy and research recommendations. The article would benefit from a clearer connection between the ecological, economic, and policy dimensions of the issue, ensuring a more comprehensive and practical contribution to the field.

6. PLOS authors have the option to publish the peer review history of their article (what does this mean?). If published, this will include your full peer review and any attached files.

Reviewer #1: **Yes: **S. A. Edalatpanah

Reviewer #2: No

---

## [Author Response · Author response to Decision Letter 0]

13 Feb 2024

Response to reviewer1

Dear Editors and Reviewers:

Many thanks for your valuable comments and suggestions on our manuscript entitled “Analysis on the control of the black tiger shrimp in the America from the perspective of international cooperation” (Manuscript ID: PONE-D-23-38997). The comments and suggestions are very helpful for improving our paper. We have made revision based on the comments and suggestions. Please find our response as follows, and we have made revision which marked in blue in the paper. Attached please find the revised version, which we would like to submit for your kind consideration.

Point 1：

Conceptual framework - While multiple control strategies are proposed, the theoretical/conceptual framework that guides comparison and analysis of the strategies is not clearly established. A stronger conceptual underpinning is needed to systematically examine the different approaches.

Response 1: 

Thank you very much for your suggestion. In the revised version, the paper establishes a theoretical/conceptual framework to guide comparative and analytical strategies, thereby establishing a stronger conceptual basis for systematically reviewing different approaches. For details, see lines 314 through 335 in blue.

Point 2：

Methodology - The methodology section does not provide sufficient detail on the differential game models constructed. Key elements like model structure, assumptions, parameters, solution concepts etc. are missing. More robust and clearly described methodology is essential.

Response 2: 

Thank you very much for your suggestion. In the revised version, this paper constructs more details of the differential game model. The research hypothesis is described in more detail in this paper. For details, see lines 355 through 359, 397 through 408 in blue. This article explains why these parameters are used. For details, see lines 411 through 421 in blue. In the revised version, the relevant parameters that are difficult to understand are explained. For details, see lines 424 through 448 in blue. This paper introduces the steps of modeling using game theory. For details, see lines 450 through 475 in blue. 

Point 3：

Literature review - Literature cited does not sufficiently synthesize previous findings on control of black tiger shrimp specifically. More focused review of related work is needed to position the current study appropriately. Also update the recent works in this field, such as:

[]Das, D. ., & Samanta, G. C. . (2024). An EOQ Model for Two Warehouse System During Lock-Down Considering Linear Time Dependent Demand. Transactions on Quantitative Finance and Beyond, 1(1), 15–28.

[]Aguirre-Pabón, J., Chasqui, L., Muñoz, E., & Narváez-Barandica, J. (2023). Multiple origins define the genetic structure of tiger shrimp Penaeus monodon in the colombian Caribbean Sea. Heliyon, 9(7).

[]Yun Tseng, H. ., Salam, N. . A. . ., Kazemi, M. ., Nadiheidari, H. ., & Mehlabani, E. G. . (2024). Reevaluating the Impact of Corporate Governance Mechanisms on Investment Management, Financing Strategies, and Corporate Performance. Management Analytics and Social Insights, 1(1), 1–16.

[]Vu, N. T., Jerry, D. R., Edmunds, R. C., Jones, D. B., & Zenger, K. R. (2023). Development of a global SNP resource for diversity, provenance, and parentage analyses on the Indo-Pacific giant black tiger shrimp (Penaeus monodon). Aquaculture, 563, 738890.

Response 3: 

Thank you very much for your suggestion.In a revised version, this paper presents a more targeted review of the relevant work to properly position the current research. At the same time, in the revised version, this article has added those references.

Point 4：

Analysis - Limited analysis is conducted to derive insights from the game models. Results and comparisons between the control strategies based on equilibrium outcomes are not clearly presented. Deeper strategic analysis is required.

Response 4: 

Thank you very much for your suggestion. In the revised version, this paper clearly presents results and comparisons between control strategies based on balanced results. For details, see lines 948 through 969 in blue.

Point 5：

Writing quality - Overall writing standard needs improvement in terms of clarity, flow, proper referencing etc. Manuscript contains repetitive information and lacks cohesion.

Response 5: 

Thank you very much for your suggestion. In the revised version, a large number of modifications have been made to the article, and the overall writing standard has been improved in terms of clarity, fluency, and proper referencing. At the same time, the repeated information in the manuscript was deleted to make the content more cohesive.

Point 6：

Policy relevance - While international cooperation is highlighted, practical policy dimensions and recommendations for partner countries are not sufficiently discussed.

Response 6: 

Thank you very much for your suggestion. In the revised version, the practical policy dimensions and recommendations of partner countries are fully discussed. For details, see lines 970 through 992 in blue.

Response to reviewer2

Dear Editors and Reviewers:

Many thanks for your valuable comments and suggestions on our manuscript entitled “Analysis on the control of the black tiger shrimp in the America from the perspective of international cooperation” (Manuscript ID: PONE-D-23-38997). The comments and suggestions are very helpful for improving our paper. We have made revision based on the comments and suggestions. Please find our response as follows, and we have made revision which marked in blue in the paper. Attached please find the revised version, which we would like to submit for your kind consideration.

Point 1：

Lines 31-33: There is some confusion here, are you referring to black tiger shrimp or Louisiana red-clawed shrimp? These are different species. The black tiger shrimp, scientifically known as Penaeus monodon, is native to Indo-Pacific, Asian, and Australian waters, not to the United States as mentioned in your introduction. This species is primarily marine, rather than freshwater, which contradicts the description in the article. Therefore, please add more sources to confirm the adaptations of this species to freshwater, or give salinity ranges based on past studies.

Response 1: 

Thank you very much for your suggestion. The original description of the black tiger shrimp is less accurate. In the revised version, this paper redescribes the origin of the black tiger shrimp, indicating that it is a shrimp living in seawater. For details, see lines 32 through 38 in blue.

Point 2：

Also, while the introduction correctly identifies the black tiger shrimp as an invasive species in the United States, further emphasis on its marine nature and the specific impacts it has on the local ecosystems, particularly in southeastern U.S. waters, could strengthen the article.

Response 2: 

Thank you very much for your suggestion. In a revised edition, the paper refocuses further on its Marine nature and its specific impacts on local ecosystems, particularly in the waters of the southeastern United States. For details, see lines 32 through 38 in blue.

Point 3：

The introduction discusses various control methods, including introducing natural predators, using the shrimp as feed, and promoting them as a food source. While these are valid points, it would be beneficial to incorporate a more detailed discussion on the FEASIBILITY and ecological consequences of these methods, especially the introduction of non-native predators, which can have unpredictable impacts on local ecosystems.

Response3: 

Thank you very much for your suggestion. In a revised edition, the paper discusses in more detail the feasibility and ecological consequences of these approaches, particularly the introduction of non-native predators, which could have unpredictable impacts on local ecosystems. For details, see lines 232 through 242 in blue.

Point 4：

Lines 153-415: The section is quite extensive which is good for replicability and understanding the approach, but it might benefit from concise summarization. Are some of the protocols already done elsewhere and could be referenced?

Response 4: 

Thank you very much for your suggestion. In the revised version, this article has added some protocols that have been done elsewhere for reference. For details, see lines 161 through 188 in blue.

Point 5：

Lines 155-177: The problem description is clear when you stated the significance of controlling black tiger shrimp, but the text could be more concise and focused on directly relating the problem to the methodology used.

Response 5: 

Thank you very much for your suggestion. In the revised version, this article states the importance of controlling black tiger shrimp more succinctly and focuses on directly linking the problem to the method used. For details, see lines 195 through 199 in blue.

Point 6：

 Lines 264-285: The hypotheses listed are critical to the study but require a more robust justification. For instance, the assumption that introduced natural enemies will not negatively impact local ecology (Line 265) is significant and needs substantial evidence or references to similar studies.

Response 6: 

Thank you very much for your suggestion. In the revised version, this article provides a stronger rationale for the assumptions listed. For example, this paper presents substantial evidence or references similar studies to illustrate the importance of the assumption that introduced natural enemies do not have a negative impact on local ecology. For details, see lines 355 through 359 in blue.

Point 7：

 Lines 318-321: The variables are well-defined, but.. it would be beneficial to explain why these specific variables were chosen and how they relate to the real-world scenario of black tiger shrimp control.

Response 7: 

Thank you very much for your suggestion. In a revised version, this paper explains why these particular variables were chosen and how they relate to the reality of the situation controlled by the black tiger shrimp. For details, see lines 412 through 421, 424 through 448 in blue.

Point 8：

Lines 322-415: The differential game model is complex and seems appropriate for the study's aims. However, there's a lack of explanation about the choice of this model over other potential models and perhaps a better explanation of the formulations.

Response 8: 

Thank you very much for your suggestion. In a revised version, this article explains the choice of this model over other possible models. For details, see lines 450 through 503 in blue.

Point 9：

 Lines 201-213, 364-367: The assessment of ecological impact, especially regarding the introduction of natural enemies, is mentioned but lacks depth, please try to reference to ecological impact studies.

Response 9: 

Thank you very much for your suggestion. In the revised version, to add depth, this paper evaluates the impact of introducing natural enemies with reference to ecological impact studies. For details, see lines 232 through 242 in blue.

Point 10：

 Lines 214-235, 236-258: While the economic aspects are considered, the ecological feasibility and long-term sustainability of the proposed solutions (feed mode and 'bring to the table' mode) are not adequately addressed. 

Response 10: 

Thank you very much for your suggestion. In the revised version, the ecological feasibility and long-term sustainability of the proposed solutions (feed model and "table" model) are elaborated in more detail. For details, see lines 265 through 277, 301 through 309 in blue.

Point 11：

 Lines 356-367, 382-391, 405-415: The use of reputation as a metric is interesting, but its operationalization and measurement in the context of ecological control are not sufficiently explained.

Response 11: 

Thank you very much for your suggestion. In the revised version, the operation and measurement of reputation in the context of ecological control is elaborated in more detail. For details, see lines 529 through 548 in blue.

Point 12：

 Lines 303-316: Ignoring feed logistics costs is a significant assumption that could affect the model's applicability in real-world scenarios. This needs justification or a discussion on potential impacts.

Response 12: 

Thank you very much for your suggestion. Ignoring feed logistics costs is an important assumption. In order to increase the applicability of the model to real-world scenarios, potential impacts are demonstrated or discussed in the modified version. For details, see lines 397 through 408 in blue.

Point 13：

 I recommend clarifying or explaining how abstract concepts like "reputation" are measured and factored into the models.

Response 13: 

Thank you very much for your suggestion. In the revised version, the operation and measurement of reputation in the context of ecological control is elaborated in more detail. For details, see lines 529 through 548 in blue.

Point 14：

 Lines 471-481: While you provide a list of parameters and their values, there is a lack of explanation regarding how these values were determined. Are they based on empirical data, expert estimation, or are they hypothetical? This needs to be clarified for the reader to understand the basis of your models.

Response 14: 

Thank you very much for your suggestion. In the revised version, this article has added an explanation of how to determine these values. Explain why they take these values. For details, see lines 679 through 725 in blue. 

Point 15：

 Lines 467-520: This part of the section attempts to provide a more detailed description of social utility changes, but the transition from theoretical models to numerical analysis seems rough, perhaps a clearer explanation of how these numerical values are linked to the earlier models would be helpful?

Response 15: 

Thank you very much for your suggestion. In the revised version, this paper explains more clearly how these values relate to the earlier model. For details, see lines 665 through 678 in blue. 

Point 16：

 Lines 487-514: This is optional but there seems to be a slight repetition of the formula for calculating social benefits under different prices for black tiger shrimp. While it's important to show these variations, the presentation could be more concise to avoid redundancy, to avoid confusing the readers.

Response 16: 

Thank you very much for your suggestion. Lines 487-514 of the original article describes the social benefits of the treatment of black tiger shrimp under conditions in the United States and partner countries.

Point 17：

 Lines 522-527: The discussion acknowledges the difference in findings compared to Yuan et al. (2021) but please provide a better explanation for why these differences exist beyond stating the variance in research objects and environments.

Response 17: 

Thank you very much for your suggestion. In the revised version, in addition to providing that there are differences in the study results, there is also a better explanation for why these differences exist, in addition to differences in the subjects and environments studied. For details, see lines 778 through 782 in blue. 

Point 18：

 Lines 528-547: The authors provide reasons for introducing natural enemies, but the discussion seems somewhat speculative. The arguments could be strengthened by including empirical evidence or case studies that support these strategies.

Response 18: 

Thank you very much for your suggestion. In a revised version, this paper strengthens the argument by including empirical evidence or case studies that support these strategies. For details, see lines 793 through 796, 802, 803, 807, 808 in blue. 

Point 19：

 Lines 534-538: The impact on ecosystem balance is mentioned, but the discussion lacks specific examples or data showing how the introduction of natural enemies has historically affected ecosystems. This leaves the argument somewhat unsubstantiated.

Response 19: 

Thank you very much for your suggestion. In the revised version, this paper discusses specific examples or data to illustrate how the introduction of natural enemies has historically affected ecosystems For details, see lines 793 through 796 in blue. 

Point 20：

 Lines 554-57

---

## [Decision Letter · Decision Letter 1]

6 Mar 2024

Analysis on the control of the black tiger shrimp in the America from the perspective of international cooperation

PONE-D-23-38997R1

Dear Dr. Wang,

We’re pleased to inform you that your manuscript has been judged scientifically suitable for publication and will be formally accepted for publication once it meets all outstanding technical requirements.

Kind regards,

Mahmoud A.O. Dawood, PhD

Academic Editor

PLOS ONE

Additional Editor Comments (optional):

Reviewers' comments:

Reviewer's Responses to Questions

**Comments to the Author**

1. If the authors have adequately addressed your comments raised in a previous round of review and you feel that this manuscript is now acceptable for publication, you may indicate that here to bypass the “Comments to the Author” section, enter your conflict of interest statement in the “Confidential to Editor” section, and submit your "Accept" recommendation.

Reviewer #1: All comments have been addressed

2. Is the manuscript technically sound, and do the data support the conclusions?

Reviewer #1: Yes

3. Has the statistical analysis been performed appropriately and rigorously? 

Reviewer #1: Yes

4. Have the authors made all data underlying the findings in their manuscript fully available?

Reviewer #1: Yes

5. Is the manuscript presented in an intelligible fashion and written in standard English?

Reviewer #1: Yes

6. Review Comments to the Author

Reviewer #1: Since this revision strengthen the manuscript "Analysis on the control of the black tiger shrimp in the America from the perspective of international cooperation" and improve its overall quality, this paper can be accepted as publication.

7. PLOS authors have the option to publish the peer review history of their article (what does this mean?). If published, this will include your full peer review and any attached files.

Reviewer #1: No

---

## [Editor Report · Acceptance letter]

10 May 2024

PONE-D-23-38997R1 

PLOS ONE

Dear Dr. Wang, 

I'm pleased to inform you that your manuscript has been deemed suitable for publication in PLOS ONE. Congratulations! Your manuscript is now being handed over to our production team.

Kind regards, 

on behalf of

Dr. Mahmoud A.O. Dawood 

Academic Editor

PLOS ONE